# Properties and dynamics of mesoscale-eddies in Fram Strait from a comparison between two high-resolution ocean-sea ice models

Claudia Wekerle[1], Tore Hattermann[2,3], Qiang Wang[1], Laura Crews[4,5], Wilken-Jon von Appen[1], and Sergey Danilov[1]

[1]Alfred-Wegener-Institut Helmholtz-Zentrum für Polar- und Meeresforschung, Bremerhaven, Germany
[2]Norwegian Polar Institute, Tromsø, Norway
[3]Energy and Climate Group, Department of Physics and Technology, The Arctic University of Tromsø, Norway
[4]School of Oceanography, University of Washington, Seattle, USA
[5]Applied Physics Laboratory, University of Washington, Seattle, USA

**Correspondence:** Claudia Wekerle (Claudia.Wekerle@awi.de)

**Abstract.** Fram Strait, the deepest gateway to the Arctic Ocean, is strongly influenced by eddy dynamics. Here we analyse the output from two eddy-resolving models (ROMS and FESOM) with around 1 km mesh resolution in Fram Strait, with focus on their representation of eddy properties and dynamics. A comparison with mooring observations shows that both models reasonably simulate hydrography and eddy kinetic energy. Despite differences in model formulation, they show relatively similar eddy properties. The eddies have a mean radius of 4.9 km and 5.6 km in ROMS and FESOM, respectively, with slightly more cyclones (ROMS: 54%, FESOM: 55%) than anticyclones. The mean lifetime of detected eddies is relatively short in both simulations (ROMS: 10 days, FESOM: 11 days), and the mean travel distance is 35 km in both models. More anticyclones are trapped in deep depressions or move toward deep locations. The two models show comparable spatial patterns of baroclinic and barotropic instability. ROMS has relatively stronger eddy intensity and baroclinic instability, possibly due to its smaller grid size, while FESOM has stronger eddy kinetic energy in the West Spitsbergen Current. Overall, the relatively good agreement between the two models strengthens our confidence in their ability to realistically represent the Fram Strait ocean dynamics, and also highlights the need for very high mesh resolution.

## 1 Introduction

Fram Strait, located between Svalbard and Greenland (Figure 1), is the deepest gateway that connects the Arctic Ocean and the North Atlantic via the Nordic Seas. Many important processes of climate relevance take place in this region. On the one hand, Atlantic Water (AW) carried northward by the West Spitsbergen Current (WSC, e.g. von Appen et al., 2016) enters the Arctic Ocean as its largest oceanic heat source. In the last decades, an increase in AW temperature has been observed in Fram Strait, with implications for the Arctic Ocean's sea ice decline (Beszczynska-Möller et al., 2012; Polyakov et al., 2012). On the other hand, a part of the AW recirculates in Fram Strait and continues southward in the East Greenland Current (EGC, e.g. de Steur et al., 2009). This water mass, which was densified on its way north to Fram Strait, contributes to the Denmark Strait overflow, which forms the dense part of the North Atlantic Deep Water, a key component of the Atlantic meridional overturning

circulation (Eldevik et al., 2009). Furthermore, cold and fresh Polar Water (PW) carried southward by the EGC is injected into the cyclonic Greenland Sea Gyre, impacting convection there (Rudels, 1995), and thus also the overflow across the Greenland Scotland Ridge.

The oceanic conditions in Fram Strait are strongly energetic. Already in the 1980's it was revealed by measurement campaigns such as the Marginal Ice Zone Experiments that eddies are abundant there (Johannessen et al., 1987; Smith et al., 1984). They play an important role in shaping the ocean circulation and hydrography, sea ice and ecosystem:

(1) Some eddies are shed from the WSC and travel westward, driving the recirculation of warm and salty AW. This was shown by mooring measurements (Schauer et al., 2004; von Appen et al., 2016) and model simulations (Hattermann et al.,
2016; Wekerle et al., 2017), which revealed high levels of eddy kinetic energy (EKE) in the WSC and along the recirculation pathway. It is found that EKE in the WSC is much stronger than in the Arctic interior (Wang et al., 2020).

(2) As AW recirculates, it subducts underneath cold and fresh Polar Water (PW) carried by the East Greenland Current (EGC). As shown by Hattermann et al. (2016), this region is characterised by negative values of vertical eddy temperature flux. Thus, eddy processes likely play an important role for the subduction of AW.

(3) Once the Return Atlantic Water (RAW) crosses (likely eddy mediated) the Northeast Greenland continental shelf break, part of it travels through a trough system towards the Northeast Greenland glaciers (Schaffer et al., 2017). An increase in its temperature might lead to the glaciers' destabilisation (Wilson et al., 2017), and it has been shown that eddy overturning is important for lifting AW onto the continental shelf in Fram Strait (Tverberg and Nøst, 2009; Cherian and Brink, 2018).

(4) Eddies play an important role for sea ice-ocean interaction. The marginal ice zone is influenced by eddies (Johannessen
et al., 1987) and submesoscale features (von Appen et al., 2018). By means of idealised model experiments, Manucharyan and Thompson (2017) showed that cyclonic eddies can trap sea ice and carry it to warm waters, leading to enhanced melting rates.

(5) Eddy and filamentary structures are important features for the marine ecosystem. Among other effects, they play an important role in transporting nutrients into the euphotic zone for phytoplankton production, and can cause stratification within days, thereby increasing light exposure for phytoplankton trapped close to the surface (Mahadevan, 2016).

Eddies can be generated through both baroclinic and barotropic instabilities (e.g. Cushman-Roisin, 1994). In the presence of horizontal density gradients and baroclinic instability, mesoscale eddies develop through the conversion of the available potential energy (APE) to EKE. Barotropic instability in contrast is associated with horizontal shear in jet-like currents, and eddies can be formed by receiving kinetic energy from the mean flow as shown for the Fram Strait by Teigen et al. (2011). Eddies can also be steered or trapped by topography, as observed for an eddy generated in the EGC (Smith et al., 1984). This
steering modulates the conversion between eddy and mean kinetic energy, which can be directed in both ways. Fram Strait, featured with its complex topography, strong lateral gradients in temperature and salinity (warm and saline AW in the eastern part, cold and fresh PW in the western part) and thus steep isopycnal slopes across the strait, strong convective events in the winter months and strong boundary currents (WSC and EGC), is thus a highly active and interesting region for studying eddy dynamics.

The Rossby radius of deformation, which characterises the spatial scale of eddies, is small in Fram Strait, around 4–6 km in summer and 3–4 km in winter (von Appen et al., 2016). Hallberg (2013) showed that in ocean models, a resolution of two

grid points per Rossby radius of deformation can be considered as threshold between "non-eddying" and "eddy-permitting" regimes, and thus higher resolution is needed for a model to be considered as "eddy-resolving". This poses problems for ocean models which typically operate on coarser grids. Recently, high resolution ocean models focused on the Fram Strait region have emerged, which perform well in reproducing the observed eddy activity (Kawasaki and Hasumi, 2016; Hattermann et al., 2016; Wekerle et al., 2017).

Given the possible sensitivity of simulations to model numerics, to the complex bottom topography and ocean currents in Fram Strait, it is not known whether the above cited models have a broad agreement on the representation of eddy dynamics in terms of eddy generation and propagation. Answering this question will not only add credence to our understanding of eddy dynamics, but also create a reference for developing parameterisations required by coarse resolution ocean models. The aim of this study is two-fold. First, we compare the output of two high-resolution, eddy-resolving ocean-sea ice models to answer the above question. We will show that there is good agreement in energy conversion that maintains eddy dynamics and in simulated eddy statistics as well, despite the fact that these models, namely ROMS (Shchepetkin and McWilliams, 2005; Budgell, 2005; Hattermann et al., 2016) and FESOM (Wang et al., 2014; Wekerle et al., 2017), differ in many aspects such as numerical discretisation, horizontal and vertical mesh resolution, parameterisations, global vs. regional configurations. Second, we explore and describe the properties of eddies in Fram Strait. We use an eddy-following approach to generate regional statistics focusing on the following questions: How are eddies spatially distributed? Are anticyclones or cyclones dominating? What is their typical size, lifetime and what are their main travel pathways?

## 2  Methods

### 2.1  Model description FESOM

Model output from the Finite-Element Sea-ice Ocean Model (FESOM) version 1.4 (Wang et al., 2014; Danilov et al., 2015) is used for eddy detection and tracking in this study. FESOM is an ocean-sea ice model which solves the hydrostatic primitive equations in the Boussinesq approximation and is discretised with the finite element method (Wang et al., 2008). In the vertical, z-levels are used. We use a global FESOM configuration that was optimised for Fram Strait with regional resolution (grid size) refined to 1 km in this area, and a coarser resolution elsewhere ($1°$ resolution throughout most of the world's oceans, 24 km resolution north of $40°$N and 4.5 km resolution in the Nordic Seas and Arctic Ocean; Wekerle et al. (2017)). By comparing with the local Rossby radius of deformation (around 3–6 km in Fram Strait, see above), this configuration can be considered as "eddy-resolving". It is forced with atmospheric reanalysis data from COREv.2 (Large and Yeager, 2008), and river runoff is taken from the interannual monthly data set provided by Dai et al. (2009). Tides are not taken into account in the FESOM configuration used here. The simulation covers the time period 2000–2009, and has daily output. In this study, we analyse model output for the years 2006–2009.

## 2.2 Model description ROMS

The second high-resolution model simulation used in this study is based on the Regional Ocean Modeling System (ROMS) (Budgell, 2005; Haidvogel et al., 2008; Shchepetkin and McWilliams, 2005, 2009) with a configuration optimised for Fram Strait and the waters around Svalbard (called S800). With 800 m x 800 m horizontal resolution, S800 is eddy resolving in Fram Strait. S800 was initialised with and forced at the ocean boundaries with daily ocean and sea ice data from a 4 km resolution pan-Arctic model called A4, together with tidal elevations from global TPXO tidal model (Egbert and Erofeeva, 2002). A4's initial state and boundary conditions were taken from monthly-averaged global reanalyses (Storkey et al., 2010). Atmospheric forcing in A4 and S800 used 6-hourly ERA-Interim reanalysis (Dee et al., 2011). A4 was initialised in 1993, and following A4 spin-up S800 was initialised in January 2005. Analyses in this paper are done for the period of 2006–2009. Model characteristics of ROMS, and also of FESOM, are summarised in Table 1. Additional information about S800, including discussions of its ability to reproduce boundary current observations in Fram Strait and along the continental slope north of Svalbard, is given in Hattermann et al. (2016), Crews et al. (2018) and Crews et al. (2019).

## 2.3 Eddy detection and tracking

Eddy detection and tracking algorithms are important tools to understand eddy properties such as their size, strength, lifetime and travel pathways. For datasets as large as the output of ocean models, automated methods need to be used. Eddy detection methods can be assigned to two categories, based either on (1) geometrical or on (2) physical characteristics of the flow field, or on a combination of both. In this study, we apply a method developed by Nencioli et al. (2010) to detect and track eddies simulated with ROMS and FESOM, which is based on the geometry of velocity vectors and thus belongs to the first category of methods. The eddy detection is based on four constraints derived from the general characteristics of velocity fields in the presence of eddies (Nencioli et al., 2010):

1. Along an east-west (EW) section, $v$ has to reverse in sign across the eddy centre and the magnitude of $v$ has to increase away from it.

2. Along a north-south (NS) section, $u$ has to reverse in sign across the eddy centre and the magnitude of $u$ has to increase away from it. The sense of rotation has to be the same as for $v$.

3. The velocity magnitude has a local minimum at the eddy centre.

4. The sign of vorticity cannot change around the eddy centre.

Two parameters, $a$ and $b$, which determine the minimum size of detectable vortices, have to be set in the algorithm. Parameter $a$ defines over how many grid points the increases in magnitude of $v$ along the EW axis and $u$ along the NS axis are checked, and its unit is "grid points". It also defines the size of detectable eddies, which is $a - 1$ grid points. Parameter $b$ defines the size (also in grid points) of the area used to find the local velocity minimum. After some sensitivity tests, we set $a = 4$ and $b = 3$, which equals the values used in the test case of Nencioli et al. (2010). Note that our mesh resolutions (800 m and 1 km

in ROMS and FESOM, respectively) are similar to theirs (1 km). Eddy boundaries around each detected centre are determined by the outermost closed contour of the stream function field, across which velocity magnitudes are still increasing in the
120 radial direction. This definition is different than the one used by Bashmachnikov et al. (2020) where the eddy boundary is approximated by the zero relative vorticity contour with a circle or an ellipse. Note that the method used in this study results in smaller eddy radii than the one used by Bashmachnikov et al. (2020).

To cross-validate our results, we also used the Okubo-Weiss criterion, which belongs to the second category of methods (Okubo, 1970; Weiss, 1991). Eddies are identified as areas where vorticity dominates over strain. More precisely, the area
where the Okubo-Weiss parameter

$$OW = \underbrace{(\partial_x u - \partial_y v)^2 + (\partial_x v + \partial_y u)^2}_{\text{normal and shear component of strain}} - \underbrace{(\partial_x v - \partial_y u)^2}_{\text{relative vorticity}} \qquad (1)$$

is below a threshold of $OW_0 = -0.2\sigma_{OW}$ with same sign of vorticity, where $\sigma_{OW}$ is the spatial standard deviation of OW, is considered as an eddy (Isern-Fontanet et al., 2006). Here $(u, v)$ is the horizontal velocity field.

After eddies are detected, eddy tracks are computed by comparing eddy centres in successive time steps. More precisely, if two eddies at successive time steps lie within a search radius and have the same sense of rotation, they form a track. The
130 eddy tracking scheme is thus sensitive to the prescribed search radius. A too small value might lead to a false splitting of the track, whereas a too large value would lead to more than one eddy within the searching area. As a first approximation, eddies are advected with the mean current. Considering a mean velocity of around 0.2 m/s (see e.g. Figure 5 in Wekerle et al. (2017)) and a daily mean model velocity field, a possible choice would be a search radius of 17 km. After performing sensitivity tests with different radii, we chose a radius of 14 km. This value reduced the number of occasions when several eddies were
135 detected in the searching area. Furthermore, eddies with a lifetime shorter than 3 days were discarded. We decided to use this threshold because the temporal resolution of the model output data is daily, and the eddy should form a track. This also helps to make sure that the eddies detected are real and not an over-detection due to uncertainties in the detection method. Eddies with a lifetime of at least three days are also required when computing the translation velocity needed to compute the eddy nonlinearity parameter (Section 4.7), for which centred differences are used.

For the eddy detection and tracking, we use daily model output for the time period 2006–2009 at 100 m depth. At this depth, the water mass lateral distribution is characterised by warm and salty AW in the eastern part of Fram Strait (in the WSC), and by cold and fresh PW in its western part (in the EGC). We decided to choose the depth of 100 m because both main water masses of the Fram Strait, AW and PW, are present at this depth (e.g. Wekerle et al., 2017, their Figure 9). In addition, we found that eddy vorticity has largest magnitudes at about 100 m depth (see section 4.6). Output from both models is interpolated to
a regular grid ($0.05°$ longitude x $0.01°$ latitude) which has approximately the same resolution as the original grids. Relative vorticity normalised by the Coriolis parameter $f$ at 100 m depth on Jan 1st 2006 is shown in Figure 2, as well as eddies detected by the Nencioli et al. (2010) method overlaid on the simulated Okubo-Weiss parameter. Note that the colour only shows the area with OW<-$0.2\sigma_{OW}$, i.e. the area considered as vortices. In both models, the relative vorticity field exhibits strong eddy activity, particularly along the pathway of the main currents, WSC and EGC, along the Yermak and Svalbard branches and
in the AW recirculation area. Apart from well defined eddies, the relative vorticity fields show lots of elongated filamentary

structures reminiscent to what was found by von Appen et al. (2018). They seem to have a smaller scale in ROMS than in FESOM.

## 2.4 Reynolds decomposition of eddy fluxes and kinetic energy

To estimate the contributions of mesoscale eddy field to the flow variability, we decompose a variable $x$ which can stand for velocity ($u$) or tracers ($c$) into a monthly mean ($\overline{x}$) and a daily-averaged fluctuating ($x'$) component, $x = \overline{x} + x'$. We derive the time-mean eddy flux of the tracer $c$ in the $u$ velocity direction from the equality $\overline{c'u'} = \overline{cu} - \overline{c}\,\overline{u}$. Similarly, time-averaged eddy kinetic energy (EKE) is computed as

$$EKE = \frac{1}{2}\left(\overline{u'^2} + \overline{v'^2}\right) = \frac{1}{2}\left(\overline{u^2} + \overline{v^2} - \overline{u}^2 - \overline{v}^2\right). \tag{2}$$

## 2.5 Energy budget

An energy budget can be obtained by taking the time-average of the momentum equation in the Boussinesq approximation, expressing velocity as $\mathbf{u} = \overline{\mathbf{u}} + \mathbf{u}'$, multiplying the equation with $\mathbf{u}'$, and time-averaging it again. This leads to a conservation equation for EKE (e.g. Olbers et al., 2012, chapter 12.2.1). The change of EKE in time is governed by the advection of eddies, energy transfer from mean kinetic energy (MKE) and available potential energy (APE) to EKE, and energy dissipation (vertical mixing and horizontal diffusion):

$$\frac{\partial \frac{1}{2}\left(\overline{u_1'^2} + \overline{u_2'^2}\right)}{\partial t} + \underbrace{\frac{\partial\left(\frac{1}{2}\overline{u_j}\,\overline{u_i'^2} + \frac{1}{2}\overline{u_j'\,u_i'^2} + \frac{1}{\rho_0}\overline{u_j'\,p'}\right)}{\partial x_j}}_{\text{Transport}}$$

$$= \underbrace{-\overline{u_j'u_i'}\frac{\overline{\partial u_i}}{\partial x_j}}_{\text{MKE}\leftrightarrow\text{EKE}} + \underbrace{\overline{w'b'}}_{\text{APE}\leftrightarrow\text{EKE}} + \underbrace{\overline{V_i'u_i'} + \overline{D_i'u_i'}}_{\text{Dissipation}}, \tag{3}$$

where $b = -\frac{g\rho}{\rho_0}$ is the buoyancy and $D_i$ and $V_i$ are horizontal and vertical dissipation terms. Cartesian tensor notation with summation convention has been used, with $i = 1, 2$ and $j = 1, 2, 3$. $u_i$ is thus the horizontal component of the velocity vector $u_j$, and $u_3 = w$ is the vertical velocity. In this study, we diagnose the first two terms on the right hand side of the equation. They are the main source terms of EKE, and can be related to barotropic and baroclinic instability.

## 3 Model assessment

For more than two decades, mooring measurements have been conducted across Fram Strait at around 79°N to monitor the exchange of water masses through this gateway (e.g. Beszczynska-Möller et al., 2012; von Appen et al., 2016; von Appen et al., 2019). To assess the overall model performance in reproducing the mean state and resolving the flow variability, we use the observed hydrography as well as the velocity field and compare the latter in terms of power density spectra (PDS) and EKE to the model results.

The two models simulated relatively similar spatial distributions of thermo-haline properties. The simulated mean temperature and salinity at 100 m depth reveal that the warm (>5°C) and narrow WSC closely follows the 1000 m isobath along the Svalbard shelf break (Figure 3). Recirculation of AW mainly occurs north of the Boreas Basin (north of 78°N). The western part of Fram Strait is characterised by cold and fresh polar outflow. The two models differ more significantly north of 80°N, with much warmer and saltier waters on the Yermak Plateau in FESOM. The front between cold PW and warm AW, indicated by 1°C and 2°C isotherms, is sharper in FESOM than in ROMS. This can also be seen in T/S diagrams (Figure 3e and f). Compared to the mooring observations across Fram Strait, both models relatively well represent the thermo-haline properties. ROMS shows a slightly cold bias which is not present in FESOM (ROMS: root mean square (rms) error of 1.28°C, FESOM: rms error of 0.49°C), and has earlier been identified to be associated with a cold bias in the A4 model that provides the inflow boundary conditions for S800 (Hattermann et al., 2016). The simulated thermo-haline properties in FESOM, particularly in central and eastern Fram Strait, are slightly too saline, whereas they are slightly too fresh in ROMS. The overall rms error in salinity is 0.26 and 0.31 in ROMS and FESOM, respectively.

For the comparison of velocity, current meter data from two moorings located in the WSC and three moorings located in the EGC (for locations see Figure 1) deployed during the time period 2006–2009 were used (von Appen et al., 2019). Time series of the $u$ and $v$ components of the velocity in the WSC and EGC were created by averaging over the two WSC and three EGC moorings, respectively. Daily averages of measured velocity at 75 m depth were calculated. Note that there are slight variations in the depth between the individual deployment years. The observed mean speed averaged over WSC and EGC moorings at 75 m depth is 0.22 m/s and 0.13 m/s respectively, while the mean speed of ROMS/FESOM at the mooring locations is 0.24/0.20 m/s and 0.16/0.12 m/s respectively.

Power density spectra of the horizontal kinetic energy from the observations and from the models were estimated via the Thomson multitaper method (Figure 4). For the WSC time series, we used linear interpolation to fill some mooring data gaps (maximum gap of 14 days). For the EGC time series, we used the time period Sep 8 2006 – Dec 31 2009 due to too long data gaps in early 2006. Spectra were computed for $u$ and $v$ components separately, summed and divided by 2. Slopes of the spectra between frequencies of 1/(14 days) and 1/(3 days) were computed by determining the median in $\log_{10}(0.05/\text{day})$ frequency steps and then fitting the slopes to those binned values. The slopes of the observations are both about -1.6 for WSC and EGC moorings, respectively, while ROMS/FESOM showed slopes of about -1.7/-2.0 and -2.4/-2.7 respectively. The difference between the models is larger at high frequency, which might be related to the fact that tides were simulated in ROMS and the models apply different atmospheric forcing. The differences between the models will be further discussed in Section 6.

Maps of the simulated EKE reveal high energy levels along the pathways of the WSC, the recirculation area and the EGC (Figure 5a and b). In both models, there is a lateral gradient from west to east, with a higher level of EKE in the eastern part of Fram Strait, the WSC region. This gradient is even more pronounced in FESOM than in ROMS. In the WSC region, FESOM shows a higher EKE level than ROMS. In the EGC, this is opposite, with a more energetic EGC in ROMS than in FESOM. This is also reflected in the power density spectra described above. A seasonal cycle of EKE in 75 m depth computed from current meter data of moorings deployed across Fram Strait is shown in Figure 5c. The highest level of EKE is reached in the winter months (January–March), and lowest values are reached in early autumn (September–November). Both models well

reproduce the observed seasonal and spatial variations of EKE (Figure 5b and c), except that the observation shows a higher

EKE level in the central Fram Strait than the models.

## 4 Eddy properties

### 4.1 Eddy spatial distribution and polarisation

During the time period 2006–2009, altogether 218,213 eddies were detected in the area 8°W–20°E/76°N–82°N in ROMS (and thus 149 eddies per day), with slightly more cyclones (54%) than anticyclones. The result is very similar in FESOM, with 55%

of the 244,811 detected eddies (168 per day) being cyclones. The tracking algorithm then revealed that these eddies belong to 30539 and 39040 tracks for ROMS and FESOM, respectively. In both simulations, the eddy density is highest in the eastern and central part of Fram Strait (Figure 6a,b). In contrast, the eddy density is low in the western part of Fram Strait and on the East Greenland continental shelf, that is, in areas covered by sea ice year-round. Comparing FESOM and ROMS, there are fewer eddies detected in that region in FESOM, which is also reflected in lower EKE values in the western part of Fram Strait

than in ROMS (Figure 5). Both models show a consistent pattern in the distribution of cyclones vs. anticyclones, which has strong regional differences (Figure 6c and d). Over the Svalbard shelf and along the East Greenland continental shelf break, cyclones are predominant. Anticyclones dominate along the main pathway of the WSC (along the 1000 m isobath), over the Yermak Plateau, and along the Svalbard branch.

### 4.2 Eddy size

In this study we compute the eddy radius as average distance from the eddy centre to the eddy boundary, which is defined by the outermost closed contour of the stream function field. Eddy properties such as their radius are determined at the locations where they are detected. In this sense, the eddy statistics are computed in a Lagrangian framework. Eddies detected in both models are relatively small, with 95%/92% of cyclones and 92%/87% of anticyclones in ROMS/FESOM having a radius below 10 km (Figure 7a). Averaged over the whole Fram Strait region, the mean/median radius for ROMS and FESOM is

4.9/4.1 and 5.6/4.7 km, respectively (Table 2). Eddies simulated in FESOM are thus slightly larger than in ROMS. The eddy radius compares well with the Rossby radius of deformation ($\sim$4–6 km in summer and smaller values in winter (von Appen et al., 2016)). This suggests that baroclinic instability is likely the main mechanism of eddy generation, which will be further investigated in Section 5. In both simulations, cyclones are slightly smaller than anticyclones (Table 2).

### 4.3 Eddy intensity

Here we take the Rossby number, the absolute value of relative vorticity divided by the Coriolis parameter $f$, as an index for the eddy intensity. A Rossby number of $\sim$1 indicates that the eddy is in cyclogeostrophic balance. The maximum value of daily mean relative vorticity within the eddy boundary is computed and averaged over all detected eddies. The mean/median intensity of eddies simulated by ROMS and FESOM is 0.4/0.36 and 0.28/0.24, respectively. Eddies simulated by FESOM

are thus weaker than eddies simulated by ROMS (see also Figure 7b and Table 2). The proportion of eddies with intensities

below 0.3 is larger for FESOM (63%) than for ROMS (38%). Cyclones are slightly more intensive (0.41±0.25 in ROMS and 0.29±0.19 in FESOM) and have a larger standard deviation than anticyclones (0.39±0.18 in ROMS and 0.27±0.16 in FESOM) (Table 2).

## 4.4    Eddy lifetime and travel distance

The duration over which eddies are continuously detected by the employed method is on average 10 and 11 days in ROMS
and FESOM, respectively (Figure 7e). 85%/82% of eddies detected in ROMS/FESOM have lifetimes below 15 days, whereas only 4%/6% of eddies detected in ROMS/FESOM have lifetimes above 30 days. Pathways of these long-living eddies will be analysed in the next section. Note that the eddy lifetime may be longer if one considers that eddies likely can exist for some time before and after being detected as an eddy by the tracking method. Also, a false splitting of the track could occur if the eddy moved relatively fast in combination with a too small searching area. In both simulations, there is no significant difference in
lifetime regarding polarisation. They are very similar regarding travel distance. On average, eddies travel around 34 and 35 km in ROMS and FESOM, respectively (Table 2). Again, there is no significant difference in travel distance regarding polarisation (Figure 7f). Compared to eddies generated e.g. in the Gulf Stream region, the lifetime of Fram Strait eddies is rather short (Kang and Curchitser, 2013).

## 4.5    Eddy pathways

Eddy pathways are investigated by focusing only on long-living eddies, e.g. eddies with lifetime of more than 30 days, and by classifying them by generation areas (Figures 8 and 9). In both simulations, eddies generated on the Svalbard shelf have very distinct travel pathways for cyclones and anticyclones, which is consistent with their distribution (Figure 6e and f). Cyclones tend to stay on the shelf, and populate the narrow Svalbard fjords. Anticyclones in contrast leave the shallow shelf area and tend to travel westward into the deep basin. As shown in Figure 7c, more cyclones (31% and 25% in ROMS and FESOM,
respectively) are detected in shallow areas with water depths less than 500 m than anticyclones (21% and 19% in ROMS and FESOM, respectively). Note that as the number of detected eddies on the East Greenland shelf is relatively small in both simulations, most eddies detected in shallow areas are located on the Svalbard shelf.

Anticyclones generated in the WSC core region, here defined approximately as the area between the 500 m and 2000 m isobaths, show longer travel pathways than cyclones. In both simulations, most of them travel westward along the recirculation
pathway north of the Molloy Deep (Hattermann et al., 2016), and some even continue southward along the East Greenland continental shelf break. Some eddies travel northward along the western rim of the Yermak plateau or recirculate around the Molloy Deep, while only few trajectories deviate westward south of 79°N in both models.

The asymmetric pathways of eddies generated on the Svalbard shelf and in the WSC core region can have dynamical reasons. As described by Cushman-Roisin (1994, Chapter 17), fluid parcels surrounding a rotating eddy are stretched when they move
to deeper waters and thus acquire relative vorticity. In contrast, when moving to shallower waters, on the flank of the eddy the surrounding fluid is squeezed and thus relative vorticity is decreased. This results in a secondary drift of the vortices, with

cyclones moving towards shallower regions and anticyclones moving to deeper regions. Morrow et al. (2004), based on satellite altimetry, showed that this dynamical reasoning can explain the diverging pathways of cyclones and anticyclones in different ocean basins. The asymmetry can also be explained by the different water masses present along the Svalbard continental shelf.

Along the Svalbard coast, the Svalbard Coastal Current transports cold and fresh waters northward, close to the salty and warm AW which is carried northward by the WSC a little offshore. The meandering between the two water masses, light water on the eastern side and denser water on the western side (roughly indicated by the 200 m isobath in Figure 6c,d) leads to the generation of cyclones on the eastern side and anti-cyclones on the western (offshore) side, which is comparable to eddy shedding along the Gulf Stream (e.g. Olson, 1991).

Tracks of long-living eddies generated in southern central Fram Strait, in particular those simulated in ROMS, show a high density of anticyclones in the Boreas Basin, the region between 0°EW–5°E, 76°N–77°N. More anticyclones appear to be trapped in this depression, a similar situation as occurring in the Lofoten Basin (Raj et al., 2016; Volkov et al., 2015). As in the case of eddies generated along the Svalbard shelf break, the clustering of anticyclones can be explained by the dynamical reason described above (anticyclones move towards the deeper basin, thus the centre of a depression). In this region, more long-living
(>30 days lifetime) eddies are generated in FESOM than in ROMS. This difference can be attributed to the different structure of the simulated mean flow and the temperature and salinity distribution (Figure 3), which is likely linked to the different model configurations (Table 1). The AW recirculation in FESOM is broader than in ROMS, so more eddies can be entrained with it.

Eddies generated in northern central Fram Strait tend to travel westward, then follow the East Greenland continental shelf break. Particularly, anticyclones travel westward between the northern rim of the Boreas Basin and the Molloy Deep, contribut-
290 ing to the AW recirculation.

Regarding eddies present in northern Fram Strait, both ROMS and FESOM show a high density along the western flank of the Yermak Plateau. Additionally, ROMS shows more long-living (>30 days lifetime) eddies west of the Plateau (Figure 6a-d) than FESOM (Figures 9 and 8). Eddies in this region have earlier been identified to occur with a different seasonality than would be expected from changes in baroclinic instability of the boundary current that explains the seasonality in eddy
occurrence along other parts of the shelf break (Crews et al., 2019). One of the many differences between the two models is the inclusion of tidal forcing in ROMS. The circulation and water mass transformations above the Yermak plateau are known to be strongly influenced by barotropic to baroclinic tidal conversion and mixing poleward of the semi-diurnal critical latitude (Fer et al., 2015), that may also explain the enhanced eddy generation in this region in ROMS. As revealed by FESOM, more cyclones tend to follow the Svalbard Branch, whereas more anticyclones tend to follow the Yermak Branch.

**4.6 Vertical structure and hydrographic properties**

We determined the vertical structure of eddies detected in 100 m depth with lifetime above 30 days by calculating relative vorticity/$f$ at the location of the eddy centres in the water column (Figure 10). In addition, temperature and salinity anomalies were calculated in the same way to study the hydrographic properties of eddies, with anomalies computed relative to the mean value for the month. This was done for eddies generated in the five different regions shown in Figure 1. Profiles of

relative vorticity are relatively similar in ROMS and FESOM, with most negative/positive (i.e. strongest vortices) values for anticyclones/cyclones generated in the WSC region and central Fram Strait.

The hydrographic conditions in regions WSC, central southern Fram Strait and Yermak/Svalbard Branch are characterised by warm and salty AW (Figure 3). These regions are temperature-stratified. Anticyclones generated there carry anomalously warm and salty and thus lighter waters and have depressed isopycnals, whereas cyclones carry anomalously cold and fresh and thus denser waters and have raised isopycnals (Figure 10). Western Fram Strait is characterised by cold and fresh PW, and is salinity stratified. The transition from a temperature stratified to a salinity stratified regime in the different regions may partly explain the difference in properties between ROMS and FESOM.

## 4.7 Eddy nonlinearity

We assessed the nonlinearity of eddies by computing the advective nonlinearity parameter $U/c$, where $U$ is the maximum rotational speed estimated as the maximum speed inside the eddy defined by the outer boundaries and $c$ is the translation speed of the eddy estimated at each point along the eddy trajectory from centered differences (Chelton et al., 2011). Eddies with a value of $U/c > 1$ can trap fluid in their interior and transport water properties, and are considered as nonlinear. In ROMS, 86% of the simulated eddies have a value of $U/c > 1$, and the percentage is quite similar in FESOM with 83% (see also the histogram of $U/c$ shown in Figure 7d). When considering long-living eddies only (lifetime>30 days), the percentage of nonlinear eddies is higher (94% and 92% in ROMS and FESOM, respectively). This is different in comparison with the global study of Chelton et al. (2011) who find all of the observed mesoscale eddies outside the tropics are nonlinear. However, they only consider long-living eddies with lifetime above 16 weeks. The most highly nonlinear eddies are found on the offshore side of the strongly meandering WSC and in the AW recirculation area (Figure 11). This indicates that ocean heat is transported from the main current into the deeper basin.

## 5 Energetics in eastern Fram Strait

We now analyse the source of EKE as simulated in ROMS and FESOM. We focus here on the eastern side of Fram Strait, which is the most energetic region (Figure 5). As described in section 2.5, the change of EKE in time is governed by the advection of eddies, energy transfer from mean kinetic energy (MKE) and eddy available potential energy (APE) to EKE, and energy dissipation. In this study we analyse only the first two terms on the right hand side of the EKE conservation equation (Eq. 3), which are the main source terms for EKE and are related to barotropic and baroclinic instability.

### 5.1 Barotropic instability

The transfer of MKE to EKE is related to barotropic instability. It can be expressed as the sum of two terms, the product of horizontal eddy Reynolds stress and horizontal mean shear, and the product of vertical eddy Reynolds stress and vertical mean shear. Strong velocity shear thus support barotropic instability. Here we consider only terms that contain horizontal derivatives, and assume that the terms with vertical derivatives play a minor role (as shown for the Gulf Stream region by Gula et al. (2015)).

In the two models, the energy conversion between MKE and EKE is directed in both ways: it shows an alternating pattern, with positive values indicating conversion from MKE to EKE and negative values indicating conversion from EKE to MKE (Figure 12a,b)[1]. The alternating pattern is very similar between the both models, with consistent locations and magnitude of positive and negative energy transfer. The energy transfer occurs mainly along the pathway of the WSC core, which is located approximately along the 500–1000 m isobaths (see also Figure 3). This is comparable to the Norwegian continental slope off the Lofoten islands, where barotropic instability is particularly important in the presence of steep bottom slopes as shown in a recent study by Fer et al. (2020). The magnitude of the depth averaged barotropic energy transfer of around $(0$–$1)$ $10^{-4}$ Wm$^{-3}$ obtained from a high-resolution ROMS simulation in Fer et al. (2020) compares well to the values in the WSC estimated from ROMS and FESOM shown in this study.

The relatively similar pattern in both models suggests that there is a strong influence of bathymetry, which determines positive and negative spots of energy conversion. A necessary condition for barotropic instability is that $\beta - \partial_{yy}\bar{u}$ vanishes within the domain, where $u = \bar{u}(y)$ is a zonal current with arbitrary meridional profile (e.g. Cushman-Roisin, 1994). The planetary potential vorticity is weak and can be ignored in polar regions, so we only consider the topographic $\beta$, with $\beta = -\frac{f}{H}\nabla H$ quantifying the change in potential vorticity across the bathymetry and $H$ and $\nabla H$ being the water depth and its horizontal gradient. A map of the topographic $\beta$ west of Svalbard reveals large values along the Svalbard shelf break (Figure 13a). We take the depth-averaged monthly mean meridional velocity $\bar{v}$ from FESOM and ROMS as an approximation of the along-stream velocity, and compute $\beta - \partial_{xx}\bar{v}$ (Figure 13b and c). Both models show a similar pattern. In many places along the Svalbard shelf break, $\beta$ is much larger than $\partial_{xx}\bar{v}$. However, in some places, e.g. at the entrance of Kongsfjorden (79°N) and Isfjorden (78°10'N) and along the 250 m isobath at around 80°N, $\beta - \partial_{xx}\bar{v}$ changes sign. These regions are characterised by positive values of energy conversion in both models, indicating active barotropic instability there.

## 5.2 Baroclinic instability

For baroclinic instability to be active, a horizontal density gradient must be present to provide available potential energy which can be converted to EKE. This transfer from APE to EKE can be expressed as the mean vertical eddy buoyancy flux (Eq. 3). In contrast to barotropic instability, the energy conversion between APE and EKE in FESOM and ROMS is directed mostly one way, with mainly positive values revealing conversion from APE to EKE (Figure 12c,d). As eastern Fram Strait is temperature-stratified, it is mainly the vertical eddy temperature flux that contributes to vertical eddy buoyancy flux (Hattermann et al., 2016, their Figure 3d). In both models, baroclinic instability is strongest between the 1000 m and 2000 m isobaths in eastern Fram Strait. The values are slightly weaker in FESOM than in ROMS. The weaker baroclinic instability in FESOM is also reflected by the fact that detected eddies are characterised by lower values of absolute relative vorticity (Figure 7b). Between the 500 m and 1000 m isobaths, both models show patches of negative vertical eddy buoyancy fluxes. Usually, those patches indicate regions where eddy fluxes interact with the sloping topography to lift dense water onto the continental shelf (Tverberg and Nøst, 2009), with upward sloping isopycnals near the seafloor that locally enhance the APE of the mean field.

---

[1]Note that there has been an error in the computation of the MKE to EKE conversion term shown in Figure 14a of Wekerle et al. (2017). Figure 12b shows the correct pattern.

A necessary condition for baroclinic instability is that the cross-stream gradient of Ertel potential vorticity (PV) changes sign with depth (e.g. Spall and Pedlosky, 2008). Ertel PV $\Pi$ is defined as

$$\Pi = (f\mathbf{k} + \nabla \times \mathbf{u}) \cdot \nabla b$$

$$= \underbrace{f\,\partial_z b}_{\text{vertical stretching}} + \underbrace{(\partial_y w - \partial_z v)\,\partial_x b + (\partial_x w - \partial_z u)\,\partial_y b}_{\text{tilting vorticity}} + \underbrace{(\partial_x v - \partial_y u)\,\partial_z b}_{\text{relative vorticity}}.$$

Here we compute $\Pi$ from simulated long-term mean velocity $\mathbf{u} = (u, v, w)$ and buoyancy $b$, and neglect the small terms containing derivatives of vertical velocity $w$. Figure 14 shows the Ertel PV and its gradient in zonal direction for two sections across the Svalbard shelf break (78°N and 78°50'N) for the FESOM simulation. The dominant term is the vertical stretching term, with a smaller contribution from the relative vorticity terms. The tilting terms are one order of magnitude smaller (Figure not shown). At both sections, the cross-stream gradient reveals a change in sign with depth, indicating that the mean current is baroclinically unstable. This is in agreement with studies by Teigen et al. (2011) and von Appen et al. (2016), and our simulated energy conversions (Figure 12c,d).

## 6   Discussion

### 6.1   Choice of the depth of 100 m for eddy detection

In this study, we chose the depth level of 100 m for eddy detection. Eddies present in the AW layer generally reach deeper than 100 m, so the eddy occurrence maps shown in Figure 6 are characteristic for deeper depths as well. In fact, an animation of daily averaged sections of velocity across Fram Strait, shown by Richter et al. (2018) (Movie S1 in the Supplement) using the same FESOM model output as this study, revealed that eddies in particular in the WSC region can reach very deep.

There may be some shallower eddies that we do not detect in 100 m depth. Shallow eddies have been observed in the Arctic Ocean Beaufort Gyre region, and are vertically confined by the strong stratification of the halocline (Zhao and Timmermans, 2015). Thus, using a shallower depth might cause us to overlook boundary current-origin eddies that do not penetrate the stratification below the mixed layer in the Basin. However, snapshots of relative vorticity close to the surface and at 100 m depth reveal a larger number of (small) eddies in 100 m depth, possibly due to strong stratification close to the surface (Figure not shown). A dedicated study of the vertical structure of eddies in Fram Strait as done by Zhao and Timmermans (2015) for the Beaufort Gyre region is required.

### 6.2   Connection between eddy occurrences and EKE

Although baroclinic instability is the main driver of mesoscale eddy variability, the connection between eddy occurrences and the APE to EKE conversion rate ($w'b'$) is very non-local. For one thing, eddies form as a result of nonlinear evolution of baroclinic instability waves and jet meanders. For another, mean circulation transports and modifies all eddy-like features, moving them away from the generation sites. Therefore, the observed pattern of eddy occurrences (Figure 6a,b) differ from the $w'b'$ distributions (Figures 5 and 12).

According to Martínez-Moreno et al. (2019), EKE can be divided into a part containing energy related to eddies and a part related to other effects such as meandering of the current. The non-negligible potential contribution of meandering to the calculated EKE fields can be seen in the maps of eddy occurrences. Along the main pathway of the WSC, which is roughly along the 1000 m isobath, the eddy occurrences are rather low in both ROMS and FESOM, whereas the EKE in both models shows a maximum along this isobath. In general, the spatial correspondence between high EKE and high eddy occurrence is not very strong. This mismatch could also be due to the fact that different individual eddies can have different levels of EKE. We do not expect to use the level of EKE to predict the number of eddies. As a result, the pattern of eddy occurrences fills the basin. The regions of high EKE and $w'b'$ are at the periphery, but they supply the perturbations that evolve into eddies.

## 6.3 Differences and similarities between observations, ROMS and FESOM

Despite their very fine resolution, ROMS and FESOM simulate a weaker variability in velocity than the observed in terms of the power density spectrum (Figure 4). This might indicate that the model resolution used is still insufficient to well resolve all the mesoscale eddies in the presence of numerical dissipation. A promising approach to reduce excessive dissipation in ocean models is the implementation of an energy backscatter scheme, which returns part of the over-dissipated energy back into the resolved flow (Jansen et al., 2015; Juricke et al., 2019). In a realistic application, Juricke et al. (2020) showed that eddy activity can be increased by a factor of 2, thereby also reducing biases in hydrography. Part of the variability revealed by the power density spectrum can also be attributed to the atmospheric forcing. Although the forcing datasets are different in the two cases, both of them are derived from relatively coarse reanalysis products (in particular, COREv.2 used in the FESOM simulation has a zonal resolution of approximately 1.875°) and may miss part of small-scale variability. A topic for further research is to clarify the importance of these factors.

A recent study by Bashmachnikov et al. (2020) compared properties of eddies detected from FESOM sea surface height fields, AVISO altimetry and spaceborne synthetic aperture radar (SAR), and revealed the difficulty of comparing model results to satellite data. The study showed that AVISO and SAR form two complementary data sets of large mesoscale eddies and of small mesoscale/submesoscale eddies, respectively. The mean FESOM eddy radius lies in-between of AVISO and SAR results. The resolution of 1 km in FESOM is thus too coarse to well resolve the submesoscale eddies detected from SAR data.

Snapshots of simulated relative vorticity (Figure 2) and the histogram of eddy intensity (Figure 7b) suggest that ROMS simulated finer and more intensive eddies and filaments. This indicates that the model effective resolution (Soufflet et al., 2016) in FESOM might be slightly lower than in ROMS. First, the grid size is slightly larger for FESOM (1 km vs 800 m for ROMS). This small difference in the grid size (20%) might matter as both numerical dissipation and explicit viscosity decrease with the grid size. In both models, biharmonic viscosity which scales with grid size cubed is applied. Second, FESOM1.4 is based on a collocated discretisation (an analog of Arakawa A-grid), whereas a staggered Arakawa C-grid is employed by ROMS. Because of pressure gradient averaging required by collocated discretisations, the effective resolution could be reduced. The collocated discretisation of FESOM also requires to use the no-slip boundary condition, which implies more dissipation along the boundary as well. Third, FESOM relies on implicit time stepping for external mode whereas ROMS uses a specially selected split-explicit method (see, e.g. Soufflet et al. (2016)) which is less dissipative. However, maps of simulated EKE and

its seasonal cycle (Figure 5) reveal that FESOM has a higher energy level in the WSC than ROMS, and in contrast, the energy level in the EGC is higher in ROMS than in FESOM. This is also reflected in the horizontal kinetic energy spectra (Figure 4). Therefore, there could be certain energy dissipation in ROMS, the source of which is not identified. This can be the case for the WSC region considering that the baroclinic energy conversion to EKE is even stronger in ROMS (Figures 12c,d). There might be other reasons for the difference in the simulated EKE in certain regions between the two models. In particular, a higher EKE level in western Fram Strait in ROMS might be related to the difference in the simulated sea ice. Sea ice could damp eddies through the ocean-ice stress.

Apart from the differences, both models do show a high similarity in eddy properties such as eddy lifetime, size, pathways and travel distance (Figures 7, 8 and 9). In addition, both models exhibit a very similar pattern in barotropic energy conversion in eastern Fram Strait (Figure 12a,b). The degree of similarity is quite surprising, given that FESOM uses z-levels in the vertical whereas ROMS relies on terrain-following coordinates, which might lead to differences in topographic steering. Topography is bi-linearly interpolated to grid points and only smoothed over the 2d-stencil of nearest vertices in FESOM. In contrast, ROMS requires a smoother bathymetry.

## 6.4 Implications for contributing to future model development

Our instability analysis indicates that the GM parameterisation (Gent and McWilliams, 1990; Griffies, 1998) traditionally used in coarse resolution climate models does not fully account for the effect of eddies. This parameterisation accounts for eddy-induced flattening of isopycnals, it thus parameterises the effect of baroclinic instability. However, our analysis shows that barotropic instability plays an important role in some regions, too. In particular, there are areas with conversion from EKE to MKE (blue patches in Figure 12a,b) indicating a strengthening of the mean flow, which is not taken into account by the GM parameterisation. Furthermore, over sloping bottom topography, the interaction of mesoscale eddies with the mean flow will be governed by a balance between the dissipation of APE and the homogenization of potential vorticity (Adcock and Marshall, 2000). Hence, it has been shown that interactions with sloping topography may locally increase the APE, e.g. by lifting dense water upward along the continental slope in Fram Strait as shown by Tverberg and Nøst (2009). Our analysis shows consistent patches of such reversed APE to EKE conversion along the Svalbard continental shelf break in both FESOM and ROMS, corroborating these theoretical considerations, which indicate that the GM parameterization that traditionally is used in coarse resolution climate models does not fully account for the effect of eddies. A similar result was shown recently in the study by Lüschow et al. (2019) investigating the vertical structure of the Atlantic deep western boundary current (DWBC). They find that below the core of the DWBC, eddy fluxes steepen isopycnals and thus feed potential energy to the mean flow, which is not represented in the GM framework.

## 7 Conclusions

Based on the results of two eddy-resolving ocean-sea ice models, ROMS and FESOM, we examined the properties and generation mechanisms of mesoscale eddies in Fram Strait. We found that the models agree with each other with respect to the

modelled circulation, hydrography and eddy characteristics. They simulate rather short-living eddies (lifetime is on average 10–11 days), with a very slight dominance of cyclones (ROMS: 54%, FESOM: 55%). Cyclones and anticyclones show very distinct travel pathways, e.g., cyclones generated on the shallow Svalbard shelf tend to stay there, whereas anticyclones tend to travel offshore into the deep basin. More anticyclones tend to be trapped in deep depressions. Mean eddy radius is 5–6 km, which compares well with the first baroclinic Rossby radius of deformation in this region. On average, eddies travel around 35 km in both models. Eddy cores are located at about 100 m depth on average. Cyclones are predominantly cold eddies, while anticyclones are predominantly warm eddies.

The models also agree on mechanisms driving eddy generation, with consistent patterns of conversions to EKE from the mean kinetic and eddy available potential energies. The small size of eddies explains why a very high (1 km or finer) resolution is needed to simulate them. The good agreement on eddy generation and properties despite the very different numerics of FESOM (unstructured horizontal grid with vertical z-levels) and ROMS (regular horizontal grid with a terrain following vertical coordinate) gives us confidence in their ability to realistically simulate eddy processes. The similarities of the simulated eddy fields also provide confidence in the eddy properties presented in this paper. Some differences between the two models are also identified in this work, including the intensity of eddies and the rates of energy conversion, which require more dedicated research to better understand the reasons.

*Author contributions.* CW, QW and SD contributed the FESOM data and analysis, TH and LC contributed the ROMS data, and WJvA contributed the observational data. All authors discussed the content of the manuscript and contributed to the interpretation of data and writing of the manuscript.

*Competing interests.* We declare that no competing interests are present.

*Acknowledgements.* This work was supported by the AWI FRAM (FRontiers in Arctic marine Monitoring) program (CW and WJvA). TH acknowledges financial support from Norwegian Research Council project 280727. QW is supported by the German Helmholtz Climate Initiative REKLIM (Regional Climate Change). LC acknowledges support from the Fram Centre "Arctic Ocean" flagship project "Mesoscale modeling of Ice, Ocean, and Ecology of the Arctic Ocean (ModOIE)" and Office of Naval Research grant number N00014-18-1-2694. This material is based upon work supported by the National Science Foundation Graduate Research Fellowship Program under Grant No. DGE-1762114. Any opinions, findings, and conclusions or recommendations expressed in this material are those of the author(s) and do not necessarily reflect the views of the National Science Foundation. The FESOM simulation was performed at the North-German Supercomputing Alliance (HLRN). The ROMS simulation was performed under the NOTUR hpc project nn9238k. We are grateful to the two anonymous reviewers and the editor for their constructive comments.

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

**Table 1.** Characteristics of the FESOM and ROMS configurations used in this study.

|  | FESOM | ROMS |
|---|---|---|
| Numerical method | finite elements | finite differences |
| Horizontal grid | A-grid (P1-P1 scheme) | C-grid |
| Vertical coordinate | z-levels | terrain following levels |
| Domain | global | regional |
| Horizontal mixing scheme | biharmonic Smagorinsky | biharmonic |
| Vertical mixing scheme | KPP | KPP |
| Tides | no | yes |

**Table 2.** Mean properties and their standard deviation (in brackets) for all eddies generated in the area $8°$W–$20°$E/$76°$N–$82°$N in the years 2006–2009 in ROMS and FESOM.

| Eddy type | Radius (km) | Abs. rel. vorticity (normalised by $f$) | Lifetime (days) | Travel distance (km) |
|---|---|---|---|---|
| *ROMS* |  |  |  |  |
| All eddies | 4.9 (2.8) | 0.40 (0.22) | 10 (14) | 34 (44) |
| Cyclones | 4.6 (2.6) | 0.41 (0.25) | 10 (12) | 33 (38) |
| Anticyclones | 5.2 (3.0) | 0.39 (0.18) | 10 (16) | 35 (51) |
| *FESOM* |  |  |  |  |
| All eddies | 5.6 (3.3) | 0.28 (0.18) | 11 (16) | 35 (44) |
| Cyclones | 5.3 (3.1) | 0.29 (0.19) | 11 (15) | 35 (43) |
| Anticyclones | 6.0 (3.4) | 0.27 (0.16) | 11 (16) | 35 (45) |

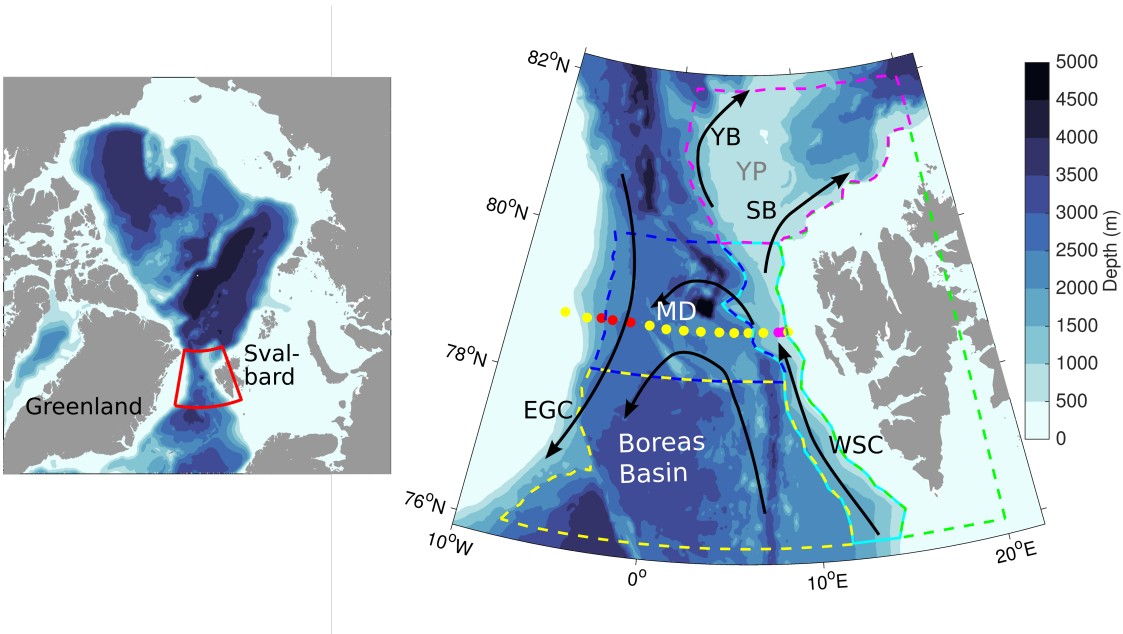

**Figure 1.** Bathymetry of the Arctic Ocean (left, red box indicates our study region) and of the Fram Strait (right). Coloured polygons in right panel indicate regions used for analysis: Svalbard shelf (green), West Spitsbergen Current (cyan), central southern Fram Strait (yellow), central northern Fram Strait (blue), and Yermak and Svalbard Branch (magenta). Coloured dots indicate moorings deployed across Fram Strait at 78°50'N; red and magenta dots show moorings used to compute velocity time series representative for the EGC and WSC, respectively. Black arrows show major currents in the Fram Strait (WSC: West Spitsbergen Current, EGC: East Greenland Current, YB: Yermak Branch, SB: Svalbard Branch). MD and YP indicate the locations of the Molloy Deep and the Yermak Plateau, respectively.

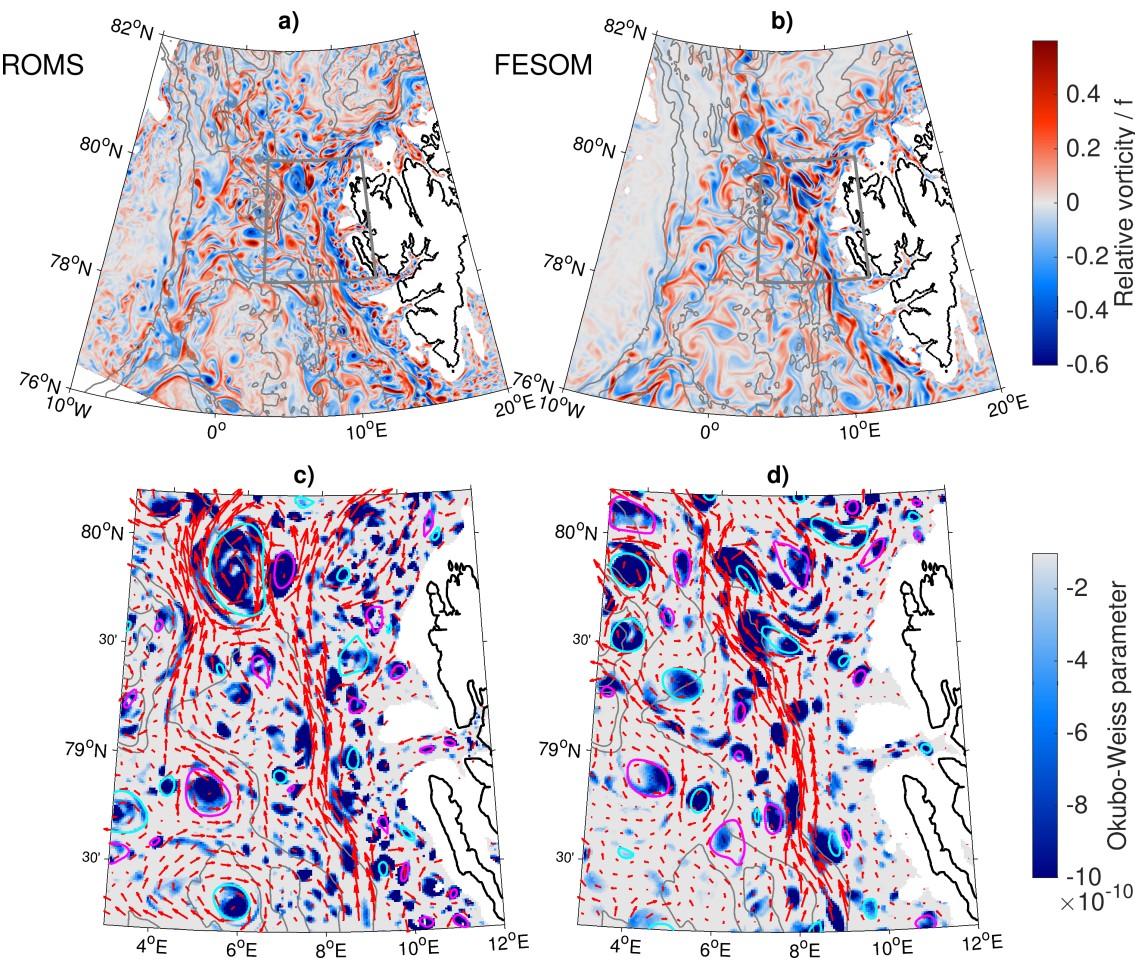

**Figure 2.** Simulated relative vorticity at 100 m depth on Jan 1st, 2006 for ROMS (a) and FESOM (b). Simulated Okubo-Weiss parameter ($s^{-2}$) at 100 m depth on Jan 1st, 2006 in a region west of Svalbard (grey box in the top panel) for ROMS (c) and FESOM (d). Shown are only values with OW<-0.2$\sigma_{OW}$, where $\sigma_{OW}$ is the spatial standard deviation of OW at that day. Red arrows show the velocity, with only every 8th vector plotted. Cyan and magenta contours show anticyclonic and cyclonic eddies respectively identified by the Nencioli algorithm (Nencioli et al., 2010). Grey contour lines indicate bathymetry at 1000 m intervals.

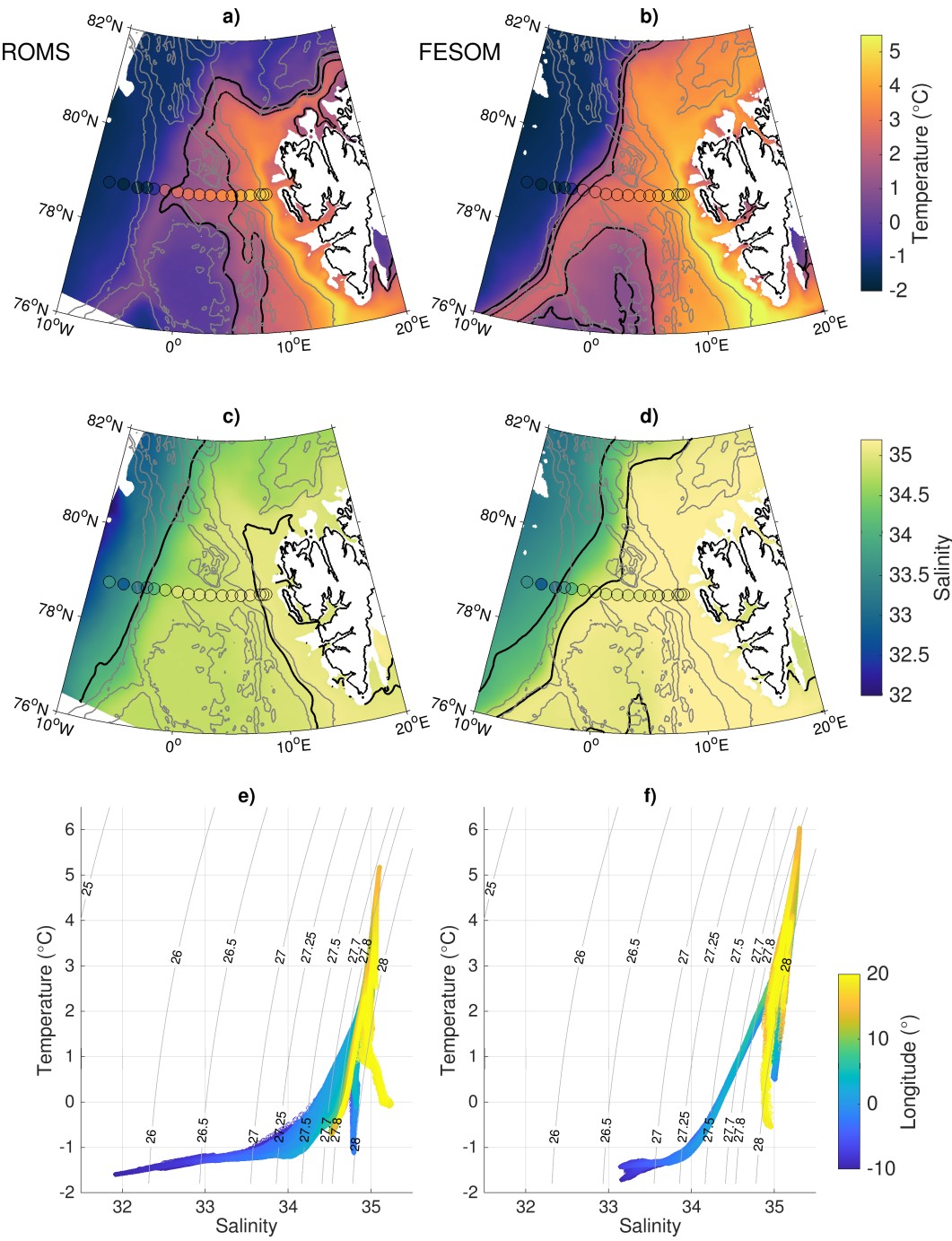

**Figure 3.** Temperature (a, b) and salinity (c, d) at 100 m depth averaged over the time period 2006–2009 simulated by ROMS (left) and FESOM (right). Black contour lines show the 1°C and 2°C isotherms and the 34 and 35 isohalines. Dots show mooring measurements in 75 m depth for the same time period (von Appen et al., 2019). Grey contour lines indicate bathymetry at 1000 m intervals. T/S diagram of simulated temperature and salinity in 100 m depth in the region 10°W–20°E / 76°N-82°N in ROMS (e) and FESOM (f). The colour shading indicates the longitude of data points.

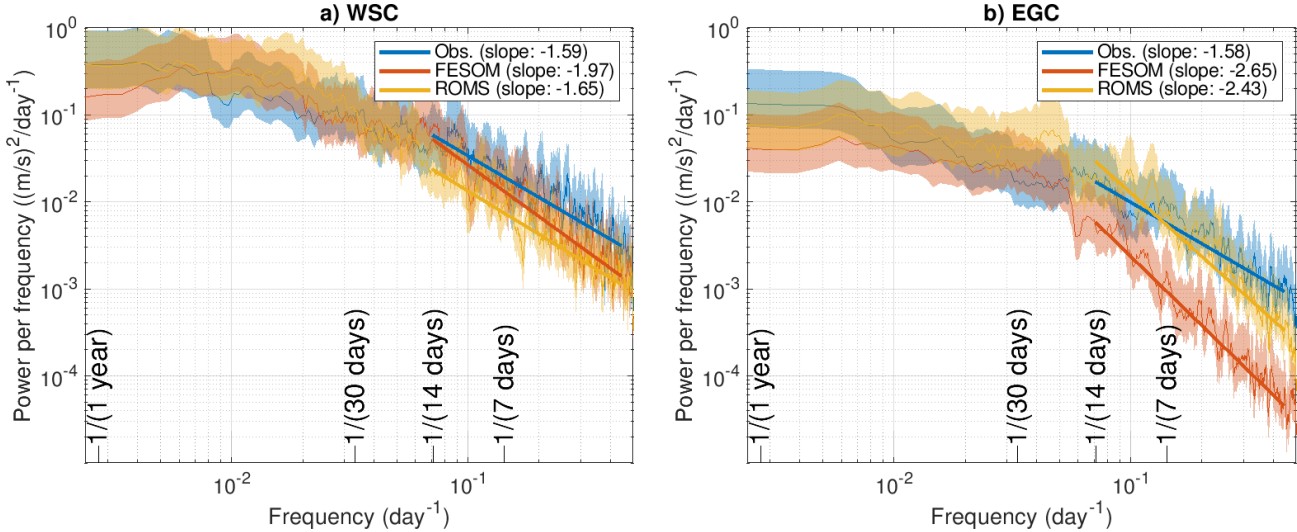

**Figure 4.** Power density spectra of horizontal kinetic energy from daily averaged velocity in 75 m depth in the (a) West Spitsbergen Current and (b) East Greenland Current from mooring measurements (blue), and models FESOM (red) and ROMS (yellow), computed as the sum of spectra of $u$ and $v$ components divided by 2. Thick lines indicate slopes of the spectra, and the shaded area indicates the 95% confidence intervals.

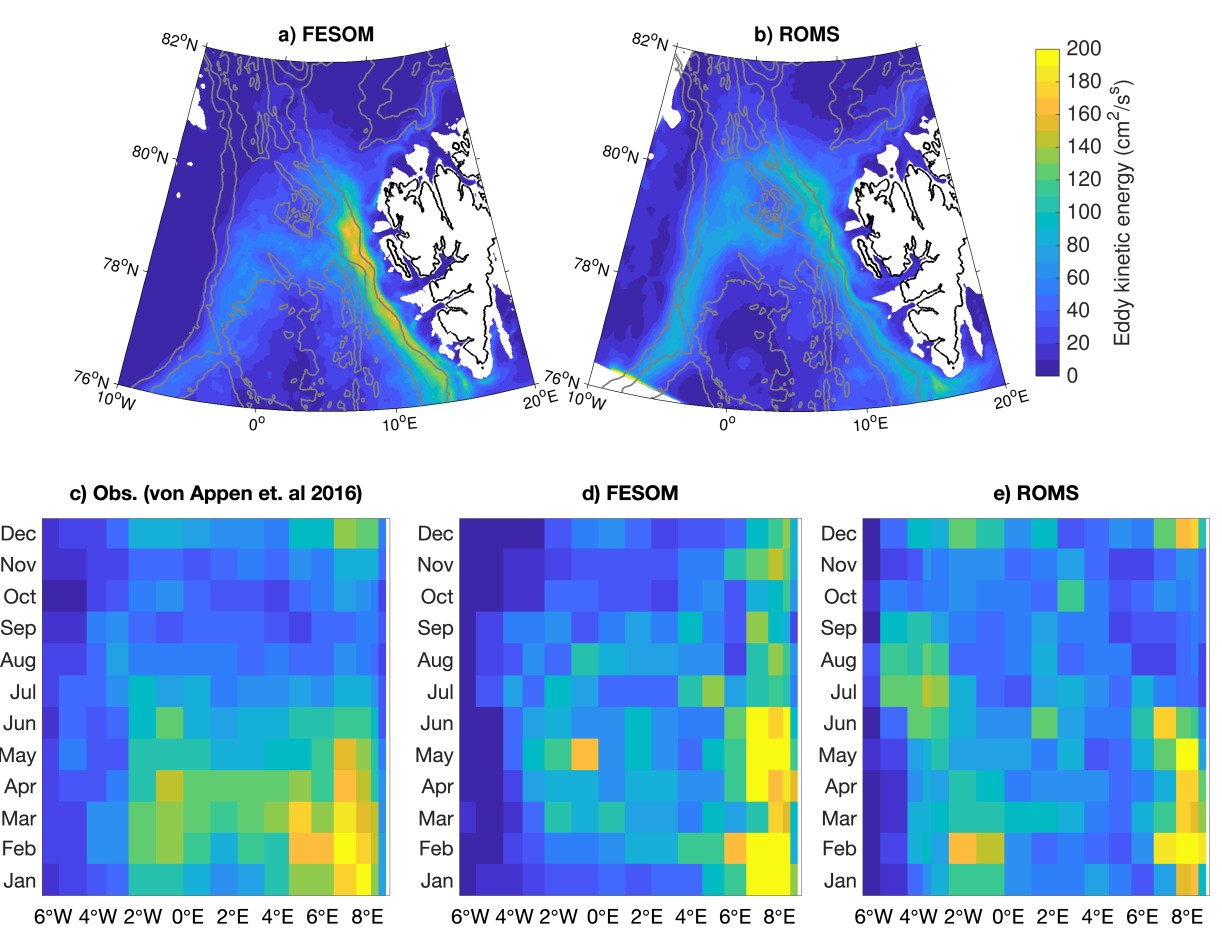

**Figure 5.** Eddy kinetic energy (EKE). Top panel: Maps of EKE at 100 m depth from (a) FESOM and (b) ROMS for the years 2006–2009. Gray contour lines indicate bathymetry at 1000 m intervals. Bottom panel: Seasonal cycle of EKE at 75 m depth across Fram Strait at 78°50'N from (a) mooring measurements (von Appen et al., 2019), (b) FESOM, and (c) ROMS for the years 2006–2009.

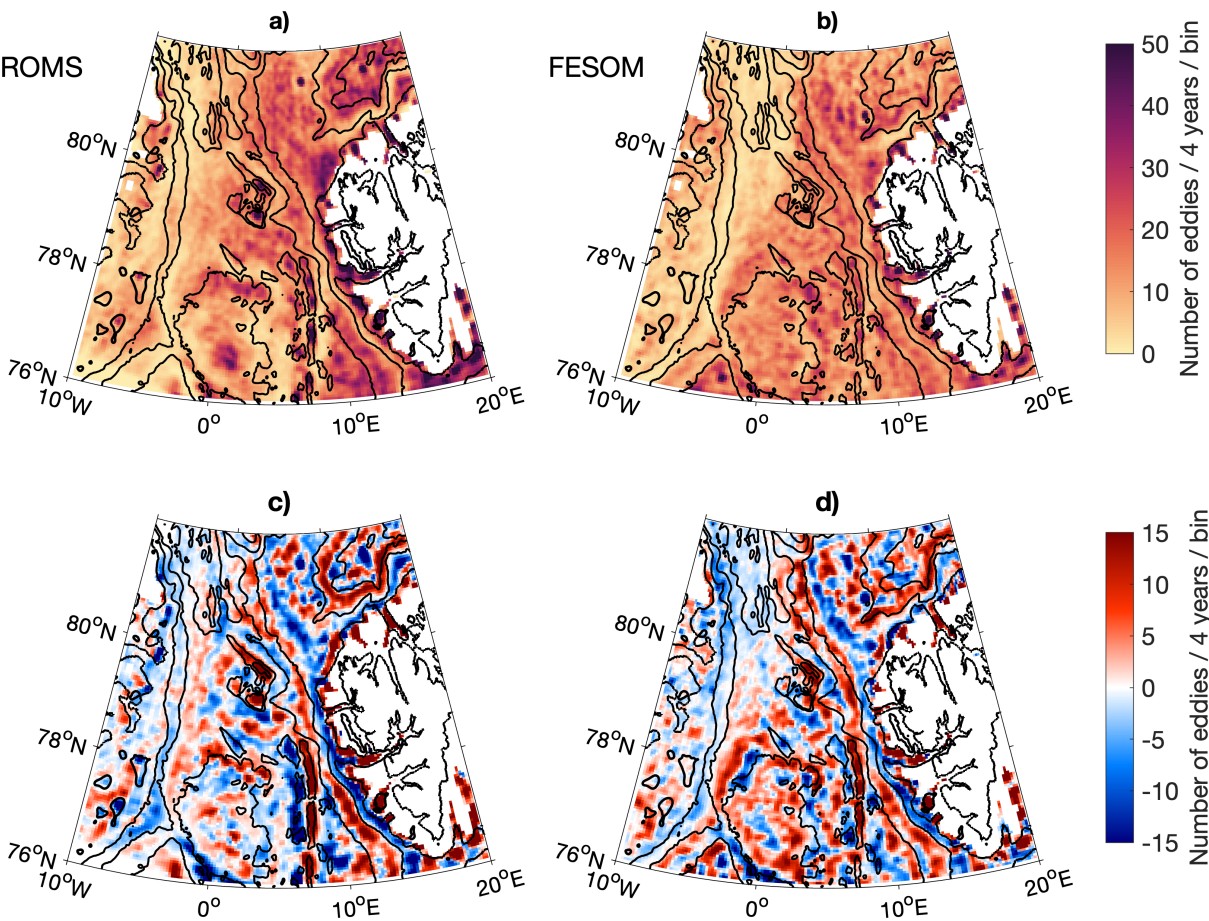

**Figure 6.** (a,b) Total number of eddy occurrences in the years 2006–2009 for (left) ROMS and (right) FESOM, binned in a 1/24° grid and smoothed with a 3 point Hanning window kernel. (c,d) Difference between numbers of cyclonic and anticyclonic eddies (cyclones minus anticyclones). Black contour lines indicate bathymetry at 1000 m intervals and at 200 m.

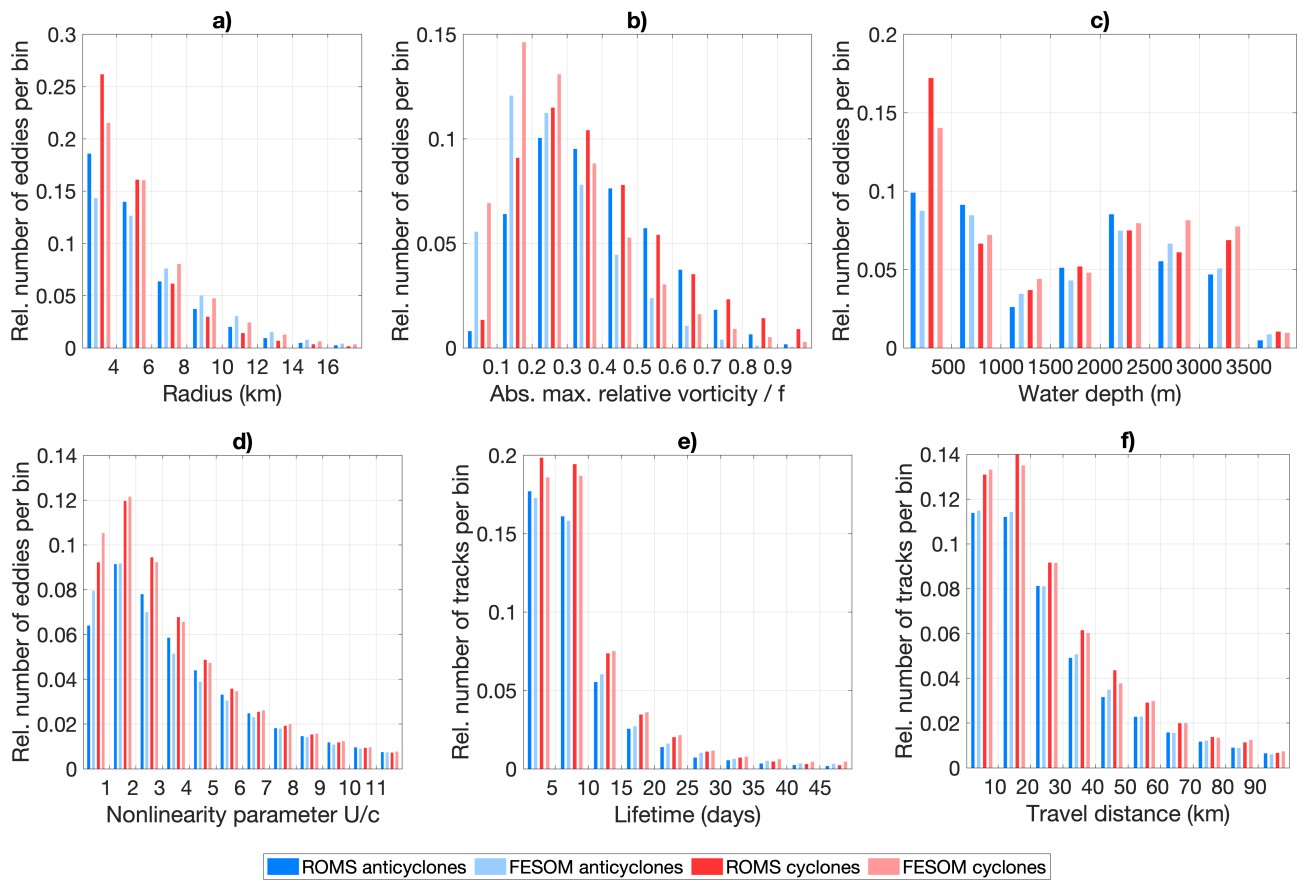

**Figure 7.** Histogram of (a) radius, (b) maximum relative vorticity normalised by $f$, (c) water depth, (d) eddy nonlinearity parameter $U/c$, (e) eddy lifetime and (f) travel distance for anticyclonic (blue) and cyclonic eddies (red) normalised by the number of eddies/tracks tracked in the area $8°W–20°E/76°N–82°N$ in the years 2006–2009 in ROMS (dark colours) and FESOM (light colours).

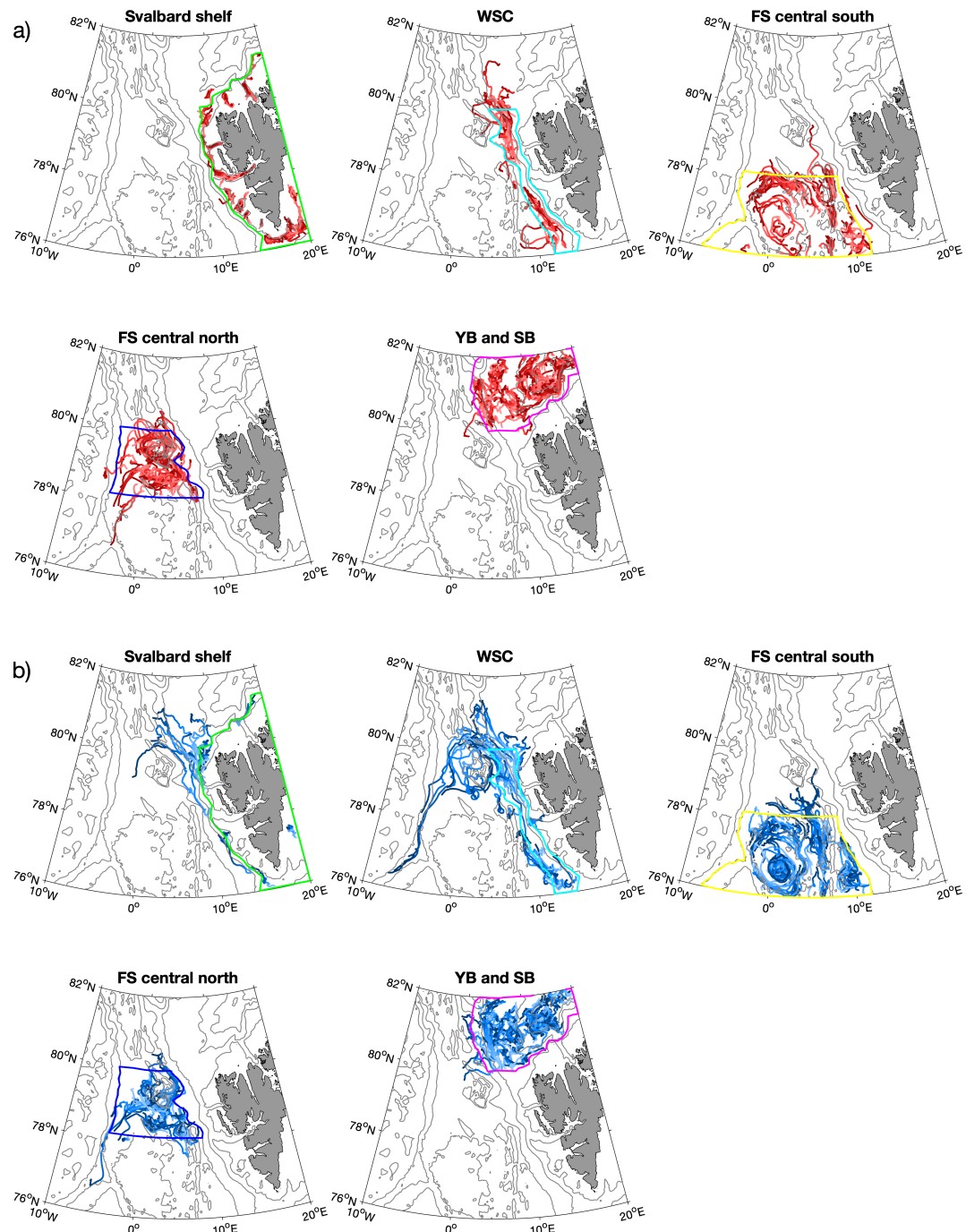

**Figure 8.** Eddy tracks of cyclones (red lines, a) and anticyclones (blue lines, b) with lifetimes of more than 30 days that are generated in five different regions indicated by coloured polygons, see Fig. 1, detected in simulation ROMS from 2006–2009. Light and dark colours of the lines indicate the beginning and end of the track, respectively.

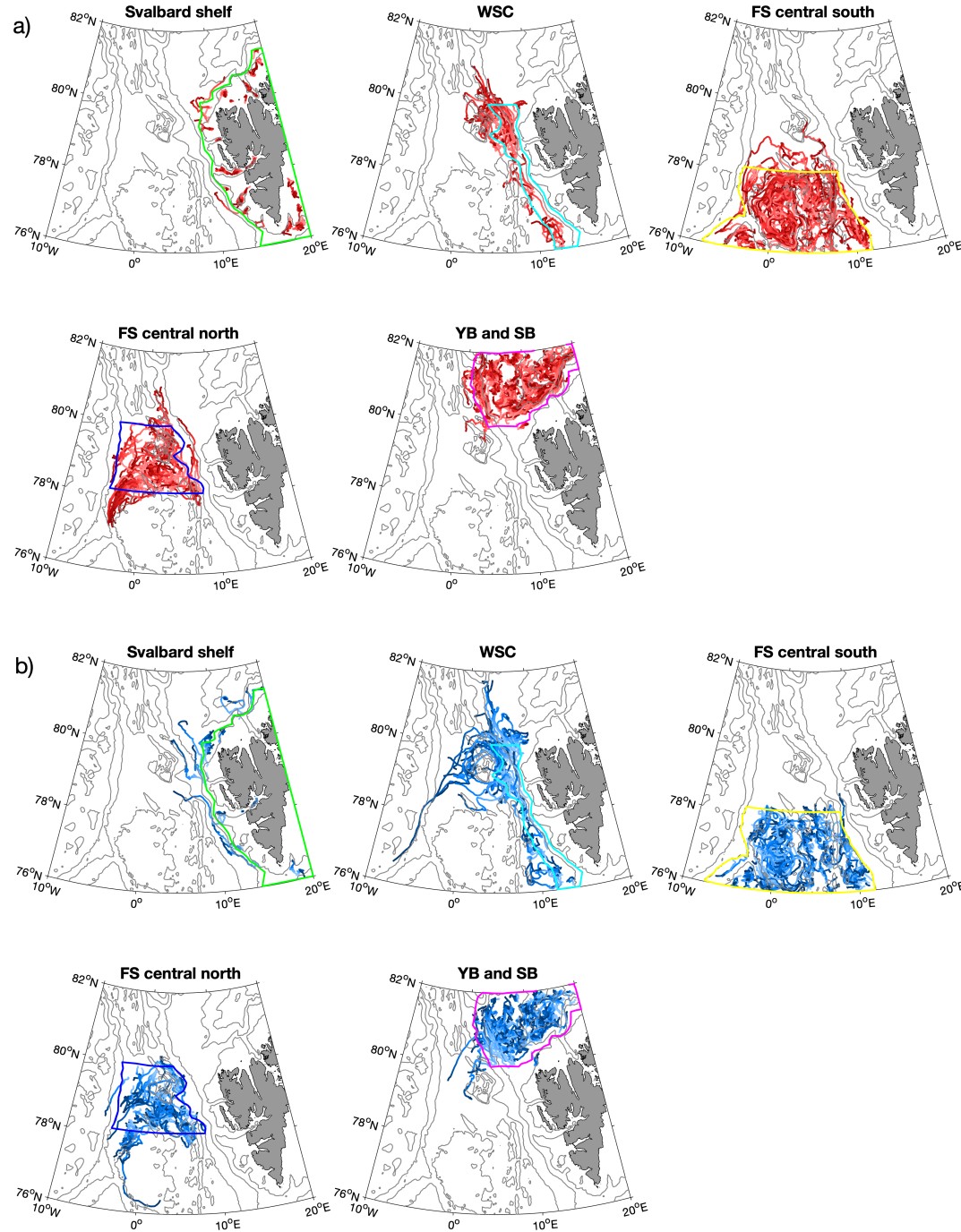

**Figure 9.** The same as Figure 8, but for FESOM.

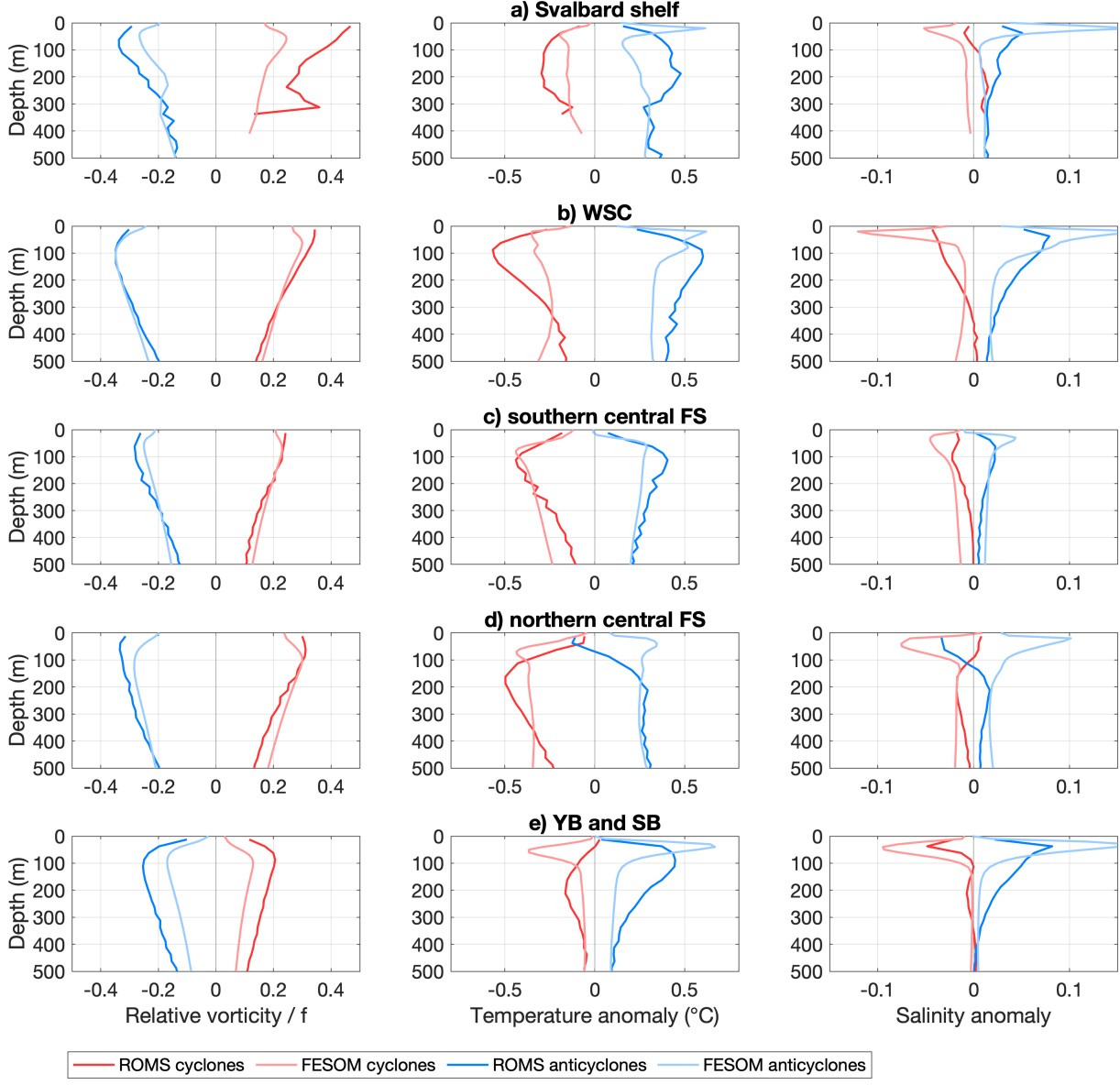

**Figure 10.** Vertical structure of eddies tracked in 100 m depth during the years 2006–2009 in ROMS (dark colours) and FESOM (light colours) with lifetimes >30 days for cyclones (red) and anticyclones (blue) generated in regions a) Svalbard shelf, b) West Spitsbergen Current, c) southern central Fram Strait, d) northern central Fram Strait and e) Yermak and Svalbard Branch. Left, middle and right panels show relative vorticity, temperature anomaly and salinity anomaly, respectively. Anomalies are calculated by taking the value in the eddy centre relative to the mean value of the month.

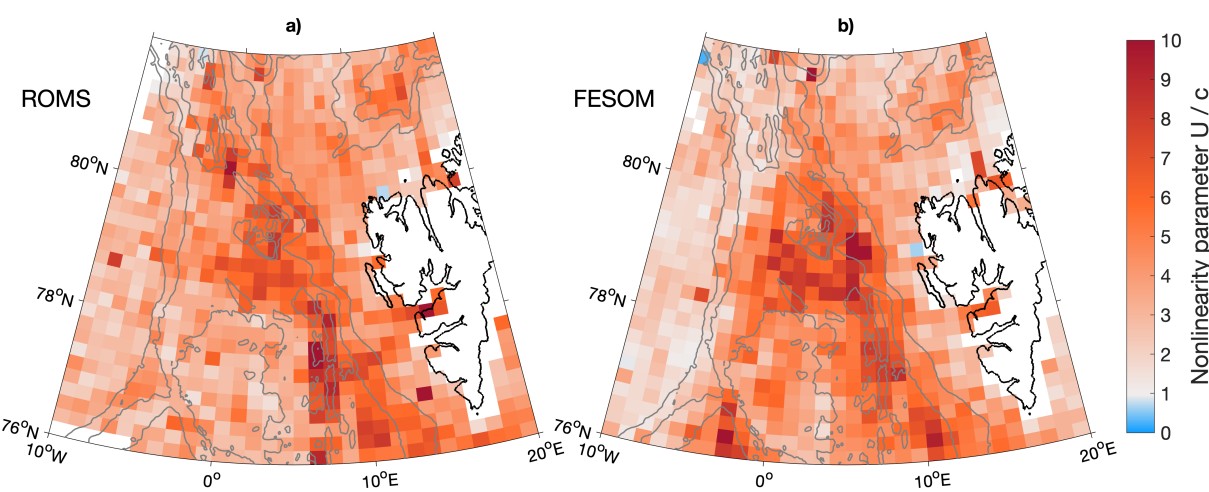

**Figure 11.** Maps of averaged nonlinearity parameter $U/c$, where $U$ and $c$ are maximum rotational and translation speeds, respectively, for eddies detected between 2006–2009 in a) ROMS and b) FESOM. Values of $U/c$ were averaged on a 1° longitude x 0.2° latitude grid. Gray contours show bathymetry contours at 1000 m intervals.

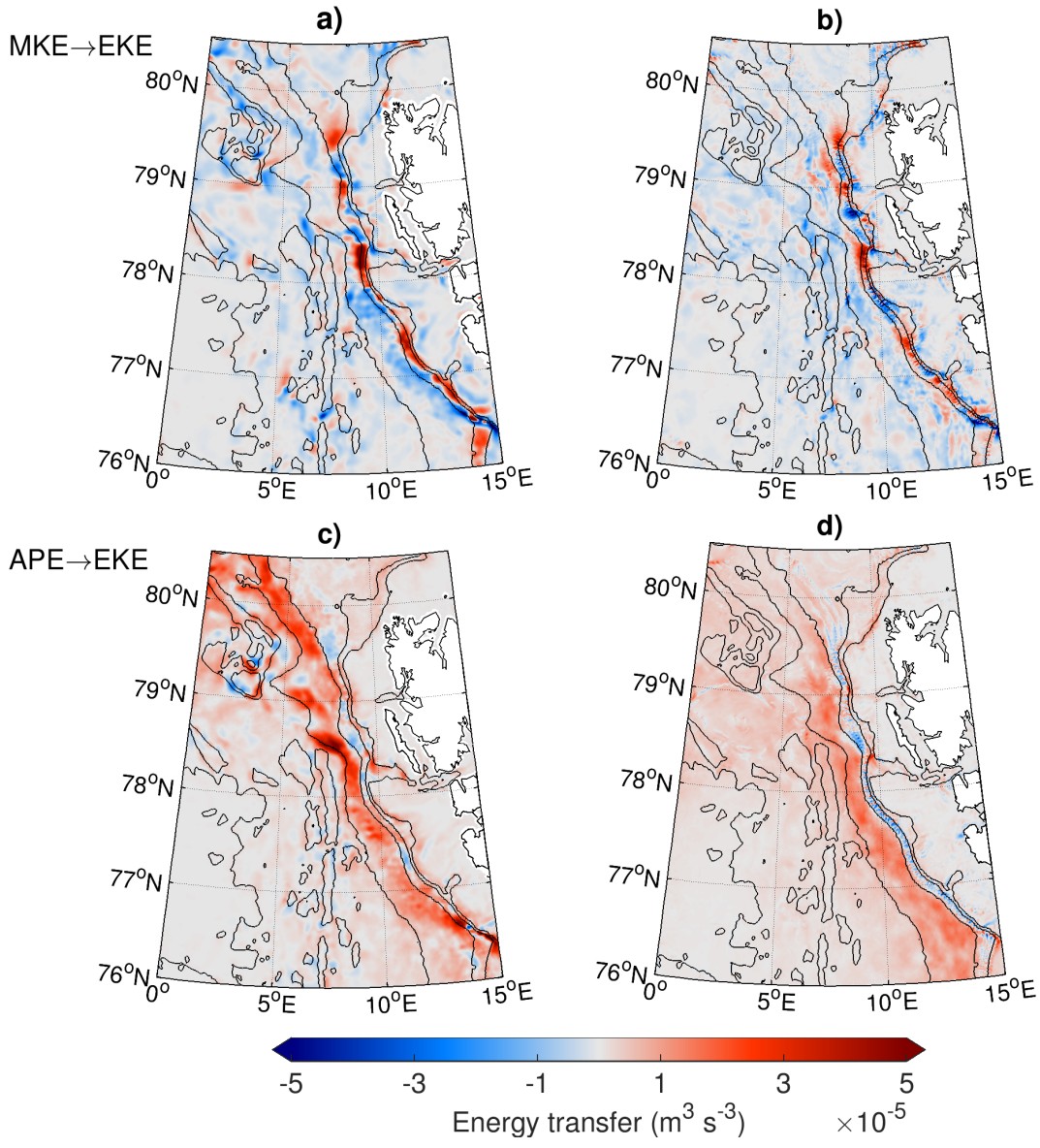

**Figure 12.** Simulated depth integrated energy transfer from (a,b) mean kinetic to eddy kinetic energy (product of horizontal Reynolds stress and mean shear, $\int_H \left( -\overline{\mathbf{u}'u'} \cdot \frac{\partial \overline{\mathbf{u}}}{\partial x} - \overline{\mathbf{u}'v'} \cdot \frac{\partial \overline{\mathbf{u}}}{\partial y} \right) dz$) and (c,d) available potential to eddy kinetic energy (vertical eddy buoyancy flux, $\int_H \overline{w'b'} dz$) averaged for 2006–2009 in (a,c) ROMS and (b,d) FESOM. Black contours show bathymetry contours at 1000 m intervals and at 200 m and 500 m depth.

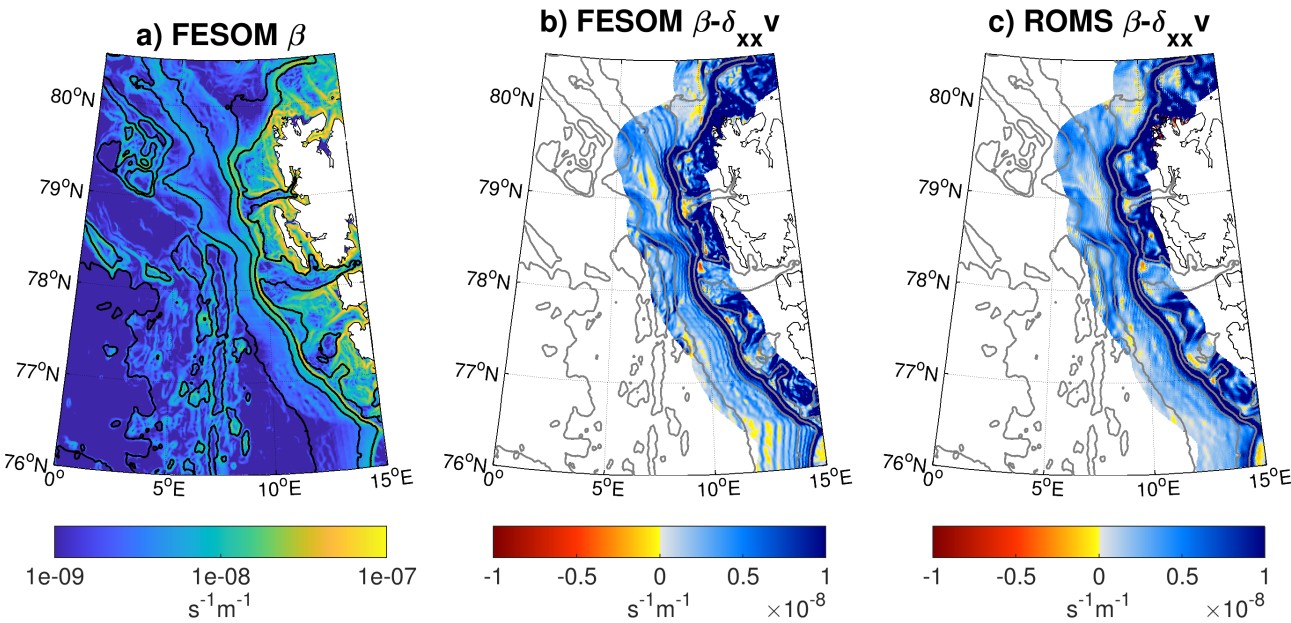

**Figure 13.** Topographic $\beta = -\frac{f}{H}\nabla H$ computed from FESOM bathymetry (a) and $\beta - \partial_{xx}\bar{v}$ for FESOM (b) and ROMS (c), where $\bar{v}$ is the simulated depth-averaged meridional velocity. The second derivative of $\bar{v}$ is computed from monthly means, and then averaged over the years 2006–2009. A change of sign of $\beta - \partial_{xx}\bar{v}$ is a necessary condition for barotropic instability. Contours show bathymetry at 1000 m intervals and at 200 m and 500 m depth. Note that values in (b) and (c) are only shown in the vicinity of the WSC main pathway (within a distance of 50 km to the 250 m isobath).

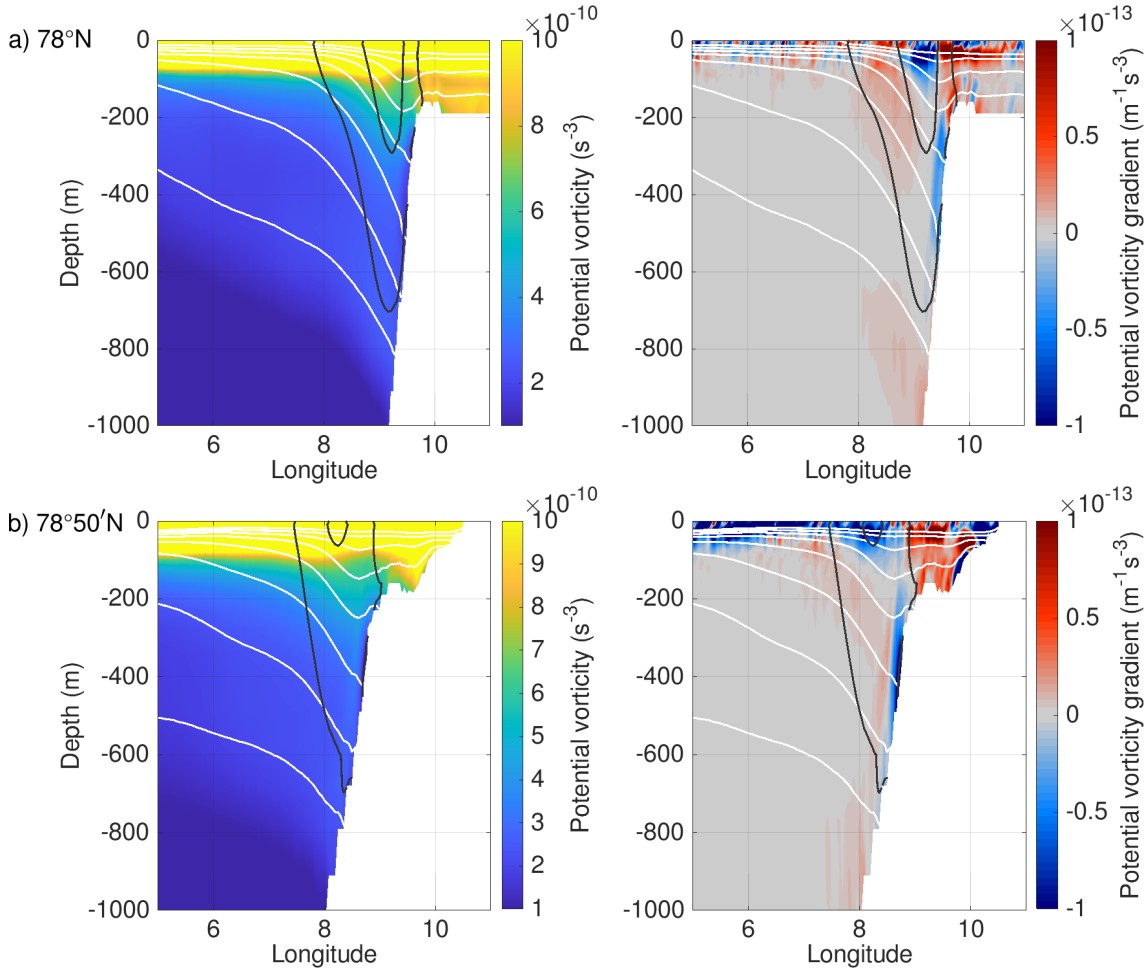

**Figure 14.** Ertel potential vorticity (left panel) and its gradient in zonal direction (right panel) across Fram Strait at (a) 78°N and (b) 78°50'N computed from long-term mean FESOM data (2006–2009). Black lines show simulated meridional velocity contours (0.1 and 0.2 m/s), and white lines show the simulated 27.9, 28, 28.1, 28.2, and 28.22 kg/m$^3$ isopycnals.