# Peer review of "Properties and dynamics of mesoscale-eddies in Fram Strait from a comparison between two high-resolution ocean-sea ice models"

_Ocean Science, 2020_

## Referee Comment (RC1) · Anonymous Referee #1 · 20 May 2020

This paper presents a comparison of eddy fields in two models: a ROMS configuration and a FESOM configuration. The models seem to agree in general but differ some in the details. The analysis focuses more on eddy statistics and there is little dynamical analysis. The main conclusion is that the two model configurations simulate roughly similar eddy statistics.

I recommend major revision because I think the paper would be much better if it demonstrated a connection between the dynamical analysis (regions of baroclinic or barotropic instability) and the eddy statistics (e.g. generation regions of long-lived eddies). The individual pieces for analyzing this connection mostly exist so this seems

reasonable to expect. Otherwise, the paper is well written and easy to understand. Some of the diagnostics used (e.g. T, S anomalies) were not described clearly — these need some major improvement (see below). The figures were nice to look at and illustrate the main points well though some extra panels would be useful (see below). Most of my other comments are minor.

1. deformation radius $\approx$ 4x grid spacing. Is this "eddy resolving" or "eddy permitting"? Some discussion with references to literature would be useful background here.

2. Line 90: Does Figure 1 show the domain for the ROMS simulation? Is this the region with refined resolution for the FESOM run?

3. Line 100: Please list the 4 constraints.

4. Line 100: Do a and b have units? What is the minimum possible detected eddy size in km? How does this compare to grid spacing?

5. Line 167: What was the longest gap you had to interpolate over?

6. Line 162: Did you average the spectra from the 3 moorings?

7. Since the eddies in FESOM are weaker (see spectra), should you adjust your criteria to be more appropriate for these weaker eddies? In other words, are you undercounting eddies in FESOM because they are weaker than the thresholds you are using?

8. Figure 6: How does this figure compare to a map of EKE averaged over the 3 years? 1. Is the sign or caption wrong in Figure 6c, d? It seems to say there are more anticyclones on the shelves contrary to line 189. 2. Also are the topographic contours the same as in previous figures? They are hard to see and it is hard to identify the 1000m isobath referred to in the text.

9. Section 4.3: This "eddy intensity" is really a Rossby number (relative vorticity / f). Can you call it that?

10. Section 4.5: These eddy pathways should be interpreted with the help of mean

velocity vectors. Jet instabilities generate cyclones on the anticyclonic side of the jet and anticyclones on the cyclonic side of the jet. For the Svalbard Branch for example, one would see anticyclones on the deeper side but cyclones in shallower side which would partially explain the asymmetry in trajectories, i.e. cyclones and anticyclones are being generated in different places. I agree that the long term tendency for anticyclones to cluster in Boreas basin is likely a topographic effect.

11. Line 255: How do you define the vertical extent?

12. Line 255: I also didn't understand how you define T, S anomalies. Are these T, S at all detected eddy center locations minus a monthly mean at the eddy center? Is this a climatological monthly mean or a monthly mean for a particular year the eddy was tracked? This definition should be clearly written.

13. Figure 8,9: Why is there a large difference in number of cyclone tracks between ROMS and FESOM in "FS Central South"

14. Lines 285-290: There are other regions with large conversions (e.g. just north and south of 77N) that are stable in Figure 12b. Would this disappear if you calculated the barotropic PV gradient properly accounting for u also? Did you use depth-averaged and monthly mean v? or monthly mean v at 100m? Why did you only use FESOM for Figure 12b?

15. "EPE" — this should really be APE (available potential energy) since eddies are deriving their KE from the APE of the /mean/ state (Vallis 2006 textbook).

16. Line 315: This is a nice demonstration that the flow is baroclincally unstable but is it unstable in the regions necessary to explain the eddy tracks in Figures 8,9? Some connection between the instability analysis and the eddy tracks is needed.

17. Lines 345, 346: I'm not sure there was any discussion of "EKE→MKE conversion associated with steepening isopycnals". Please describe this in more detail. The "energy backscatter" parameterizations being currently developed would be useful literature to reference (e.g.Jansen et al 2015)

18. Discuss sensitivity of eddy maps to choice of 100m depth

19. Figure 1: Showing a second panel with a larger region would be nice for readers not familiar with this place.

20. Figure 4: It is more conventional to show logarithm axis scales instead of showing log10(quantity) on a linear scale. Why did you choose to not do that? Please mark 1/30 days since that's the frequency you chose to separate "mean" and "eddies"

---

## Referee Comment (RC2) · Anonymous Referee #2 · 26 May 2020

Review Comments:

The present paper reports results from a sensitivity study on the ability of two different numerical circulation models (ROMS and FESOM) in reproducing comparable eddy dynamic properties in terms of generation sites and propagation pathway in Fram Strait, a region characterized by a complex bathymetric configuration. Main difference between these model lies on their numerics and formulations, which includes numerical grid discretization, horizontal and vertical mesh resolution, parameterizations, and coverage (i.e. global vs. regional models).

The study is generally well written, and the analysis performed are appropriate. The

figures are neat and clean. The discussion of the results is superficial. The authors should inspect more thoroughly the results presented. Nevertheless, the study stands as a good contribution, as it adds knowledge to the region. However, unfortunately it doesn't add new knowledge to the subject field whether in ocean modelling or in geophysical fluid dynamics. If it does, then perhaps it has not been clearly presented.

Find below few/minor points which the authors should address before its acceptance for publication.

The reference for ROMS in the introduction section may not be the most appropriate here!

Line 105 the sentence seems incomplete. In part it reads as "we also the Okubo-Weiss". Do you mean "we also used the Okubo-Weiss"?

Are there any criteria on the choice of the number of days taken as threshold to discard eddies with lifetime lesser than 3 days? Please note that the caption in Figure 10 and in other parts of the manuscript indicate a lifetime of 30 days. Is there any inconsistency??

Paragraph 130. What do you mean by eddy detected by experts?? Please make it clear.

Did you use any filtering on the field of Okubo-Weiss parameter?

Perhaps the differences between the model maps shown in Figures 2 and 3 should be quantified to highlight geographical/spatial sites where the models converge and where they diverge (e.g: r=ROMS – FESOM for the parameters presented in Fig2 and Fig3).

Under the Model assessment section, paragraph 155, the authors indicate the model simulates similar spatial distribution of the water masses. However, no T/S diagrams have been shown. Perhaps replacing the term "water-masses" with "thermo-haline properties" would be more appropriate. The same is true wherever this term appears in this manuscript.

[Figure]

Lines 170 – 173, is the difference only related to the effects of tides simulated in ROMS? Could not be also somehow related to differences in the surface forcing fields between the models?

Lines 180 – 184: Are the eddy statistics computed in a Lagrangian or Eulerian frame of reference? Please clarify this important aspect.

An important eddy property which could be included in this study is the eddy nonlinearity parameter. This parameter would give a good insight on the eddy's coherence and ability to trap and transport material along their pathway of propagation.

Lines 248 – 249. Are the tides the only difference between the models? What about the vertical discretization of the water column??

In section vertical extension and hydrographic properties: Are the values of vorticity in the eddy centers, a single point values or are averaged values within the eddies? Please make this aspect clear.

Line 279: is the superscript number after (Figure 11a, b)1 meant to be there?

---

## Editor Comment (EC1) · Ilker Fer (Editor) · 20 Jun 2020

Dear Claudia,

Thank you for this interesting manuscript. As you see, both reviewers have some constructive comments. I wanted to drop this editor comment so that you can consider the points below when preparing your final response and the revised version. My major criticism is that the discussion section is not well developed and must be improved.

Here is a list of minor issues which must be clarified or written differently:

Li 25: "strongly turbulent": presence of mesoscale eddies does not make the oceanic

conditions strongly turbulent. If this is supported by microstructure measurements, please use and cite; if not please choose a different wording (energetic?).

Li 33: do eddies lead to vertical eddy fluxes? I thought they would lead to lateral fluxes. Otherwise, please clarify the pathway from lateral to vertical.

Li 38. How is the MIZ "shaped" by eddies?

Li 47-48: I do not think Teigen et al or Johannessen et al are the key references for the theory or dynamics related to barotropic instability or topographic steering/trapping of eddies, respectively.

Li 92: Sundfjord et al (2017) is about a Svalbard fjord and I am not sure how it is relevant to the ability of the model to reproduce the slope boundary current.

Li 100: What are "a and b". Please clarify.

Li 105: we also use

Li 117 and 118: please justify the choices of 3 day and 100 m depth

Li 130: by experts? (please clarify)

Heading 2.4, remove one "and"

Li 145: these terms do not indicate instability but rather conversion from MKE to EKE and EPE to EKE (which can be related to the instabilities you mention).

Li 169-171: the syntax of frequencies and slopes are difficult to follow for a reader

Li 193: did you introduce a stream function?

Li 200: Isn't this the Rossby number?

Li 229: Molloy

Li 293: The Ghaffari analysis is from a 2-layer model? I am not sure how this is directly comparable (at least you might want to point this out). About the instability of

the slope current along the Lofoten escarpment, please see some recent conversion rate calculations similar to yours, using high resolution ROMS fields (Section 9, in a otherwise mooring observation paper): Ocean Sci., https://doi.org/10.5194/os-16-685-2020. Note that I authored this paper, so feel free to ignore this suggestion. However, the conversion rate fields (Fig 11 in both papers) are directly comparable.

Li 309: Need a dot product before the buoyancy gradient?

Discussion section is not appropriate and must be improved. Also I note that the last paragraph (on providing information to develop GM type parameterization) is not really supported by your results or built upon them in a convincing way. Please improve this part or remove.

Opening paragraph of the Conclusions is not conclusions (or findings from your study), and could be integrated to discussion or removed.

Fram Strait: (If I'm not wrong) in the English usage, you should drop "the" in front of Fram Strait (except referring to a specific feature associated with Fram Strait, say, the Fram Strait circulation etc.) This must be corrected throughout, including the title.

Fig 4: I think it is not very meaningful to show the spectra of daily averaged speed (this is what you mean by absolute velocity?). It would be better to show the sum of spectra from u and v components (this corresponds to distribution of double the horizontal kinetic energy, or divide by two and call it HKE spectrum). In any case, you need units on y axis ((m/s)^2/(1/day) ?). It would be helpful with 95% confidence intervals on the spectra.

Thank you —————-

---

## Author Comment (AC1) · 15 Jul 2020

**Reviewer 1**

This paper presents a comparison of eddy fields in two models: a ROMS configuration and a FESOM configuration. The models seem to agree in general but differ some in the details. The analysis focuses more on eddy statistics and there is little dynamical analysis. The main conclusion is that the two model configurations simulate roughly similar eddy statistics.

I recommend major revision because I think the paper would be much better if it demonstrated a connection between the dynamical analysis (regions of baroclinic or barotropic instability) and the eddy statistics (e.g. generation regions of long-lived eddies). The individual pieces for analyzing this connection mostly exist so this seems reasonable to expect. Otherwise, the paper is well written and easy to understand. Some of the diagnostics used (e.g. T, S anomalies) were not described clearly —these need some major improvement (see below). The figures were nice to look at and illustrate the main points well though some extra panels would be useful (see below). Most of my other comments are minor.

We would like to thank you for the constructive comments, which significantly improved the manuscript. A major comment is about the connection between eddy statistics and dynamics. However, there is no 1:1 correspondence between (a) sites of eddy generation through local instability, (b) the energy conversion as a result of the presence and interaction of an eddy field with the background mean state and (c) the statistics of the diagnosed eddy trajectories (only considering coherent vortices, but not all the filaments and eddy genesis in-between). In the revised version, we discuss the connection between these points (Section 6.1). A deeper analysis, e.g. computing unstable growth rates as done by Isachsen (2015, https://doi.org/10.1002/2014JC010448), would be out of the scope of this paper.

As suggested, we improved the description of the diagnostics. Below are detailed answers to your comments.

1. deformation radius≈4x grid spacing. Is this "eddy resolving" or "eddy permitting"? Some discussion with references to literature would be useful background here.

We added this sentence in the introduction:
"Hallberg (2013) showed that in ocean models, a resolution of two grid points per Rossby radius of deformation can be considered as threshold between "non-eddying" and "eddy-permitting" regimes, and thus higher resolution is needed for a model to be considered as "eddy-resolving"."

2. Line 90: Does Figure 1 show the domain for the ROMS simulation? Is this the region with refined resolution for the FESOM run?

As you also suggested in 19., we added a second panel in Figure 1 showing a map of the Arctic Ocean. The red box indicates the area (76°N-82°N, 10°W-20°E) used for eddy detection. This region comprises the domain of the ROMS simulation (except for very small parts in the north-west and south-west corners, which can be seen in Fig. 2a). It is also the region with refined resolution in FESOM.

3. Line 100: Please list the 4 constraints.

We now listed the 4 constraints in section 2.3.

4. Line 100: Do a and b have units? What is the minimum possible detected eddy size in km? How does this compare to grid spacing?

We added a better description of a and b in section 2.3:
"Parameter a defines over how many grid points the increases in magnitude of v along the EW axes and u along the NS axes are checked, and its unit is grid points. It also defines the minimum size of detectable eddies, which is a-1 grid points. Parameter b defines the size (also in grid points) of the area used to find the local minimum of velocity."

5. Line 167: What was the longest gap you had to interpolate over?

For the WSC time series, the longest gap was 14 days. For the EGC time series, we used the time period Sep 8 2006 - Dec 31 2009 to avoid a long gap in the beginning of 2006. We added this description in the text.

6. Line 162: Did you average the spectra from the 3 moorings?

Yes, we averaged the velocity components u and v over the moorings, and then computed the spectrum from this time series. We added this clarification.

7. Since the eddies in FESOM are weaker (see spectra), should you adjust your criteria to be more appropriate for these weaker eddies? In other words, are you undercounting eddies in FESOM because they are weaker than the thresholds you are using?

We added a new panel in Figure 5 showing a spatial map of eddy kinetic energy (EKE) in both models. As also visible from the seasonal cycle of EKE, FESOM shows a higher

energy level in the WSC than ROMS, and in contrast, the energy level in the EGC is higher in ROMS than in FESOM. This is also reflected in the spectra.
Even if eddies were weaker in FESOM, they are defined by a circular velocity structure which is detected by the algorithm. Thus, the Nencioli method does not depend on the strength of the velocity field, but on the geometry of the flow field (closed contours). Both models have the same resolution (interpolated to the same regular grid), so there is no risk of undercounting.

8. Figure 6: How does this figure compare to a map of EKE averaged over the 3 years? 1. Is the sign or caption wrong in Figure 6c, d? It seems to say there are more anticyclones on the shelves contrary to line 189. 2. Also are the topographic contours the same as in previous figures? They are hard to see and it is hard to identify the1000m isobath referred to in the text.

We added maps of EKE averaged over 2006-2009 to Figure 5. The comparison of the EKE maps and eddy occurrences is discussed now in section 6.1. In fact, Figure 6c,d shows that there are more cyclones on the shelves than anti-cyclones (red color, where bottom topography is shallower than 200 m). We changed the topographic contours and increased the linewidth.

9. Section 4.3: This "eddy intensity" is really a Rossby number (relative vorticity / f). Can you call it that?

Yes, you are right. We take the Rossby number as an index for the eddy intensity. We added the description in section 4.3. This was also done by Kang et al 2014 (https://doi.org/10.1002/jgrc.20318) in their study of Gulf Stream eddies.

10. Section 4.5: These eddy pathways should be interpreted with the help of mean velocity vectors. Jet instabilities generate cyclones on the anticyclonic side of the jet and anticyclones on the cyclonic side of the jet. For the Svalbard Branch for example, one would see anticyclones on the deeper side but cyclones in shallower side which would partially explain the asymmetry in trajectories, i.e. cyclones and anticyclones are being generated in different places. I agree that the long term tendency for anticyclones to cluster in Boreas basin is likely a topographic effect.

Thank you for this explanation. We included an extended version of explanation that resorts to the fact that there is a front between different water masses: "Along the Svalbard coast, the Svalbard Coastal Current transports cold and fresh waters northward, close to the salty and warm AW which is carried northward by the WSC a little offshore. The meandering between the two water masses, light water on the eastern side and

denser water on the western side (roughly indicated by the 200 m isobath) leads to the generation of cyclones on the eastern side and anti-cyclones on the western (offshore) side."
The situation is comparable to the Gulf Stream, where dense (cold) water is located on the northern side of the main path, and light (warm) water is located south of the main path. This leads to the generation of cyclones on the southern side, and anticyclones on the northern side of the main path. We added this description in section 4.5.

11. Line 255: How do you define the vertical extent?

For every eddy center location (of the eddy detected in 100 m depth), we compute relative vorticity, temperature and salinity in the whole water column at that location, i.e. in every model layer. We added this description in the text: "...by calculating relative vorticity/f at the location of the eddy centres..."

12. Line 255: I also didn't understand how you define T, S anomalies. Are these T, S at all detected eddy center locations minus a monthly mean at the eddy center? Is this a climatological monthly mean or a monthly mean for a particular year the eddy was tracked? This definition should be clearly written.

Right, we compute T and S at the detected eddy center locations, and then subtract the monthly mean value at this location of the particular month. Description added in section 4.6.

13. Figure 8,9: Why is there a large difference in number of cyclone tracks between ROMS and FESOM in "FS Central South"

Indeed, more cyclones are generated in FESOM in "FS Central South" than in ROMS (269 and 107 generated cyclones in FESOM and ROMS, respectively, during the years 2006-2009). Not only cyclones, but also anticyclones are more abundant in FESOM than in ROMS in this region. This is consistent with the fact that baroclinic energy conversion in this region is stronger in FESOM (Fig. 12).

14. Lines 285-290: There are other regions with large conversions (e.g. just north and south of 77N) that are stable in Figure 12b.
· Would this disappear if you calculated the barotropic PV gradient properly accounting for u also?
We computed the barotropic PV gradient by also taking into account u. In particular, we defined segments of length 0.25° along the 1000 m isobath and approximated the along-stream velocity as the velocity tangential to the segments. The across-stream derivative

was computed in the normal direction of the segments. However, the difference to the term $d^2v/dx^2$ is only minor, and does not reveal visible difference.

Nonetheless, there are small regions where the barotropic PV gradient changes sign near 77°N, which are maybe hard to see in Figure 12b. We modified the colormap slightly, and hope that the figure is more clear now.

·   Did you use depth-averaged and monthly mean v? or monthly mean v at 100m?

We used depth-averaged monthly mean v velocity. We added this description in section 5.1.

·   Why did you only use FESOM for Figure 12b?

Since the energy transfer terms in Figure 11a,b are very similar, we do not expect significant differences. For simplicity and also practically to reduce the workload of coauthors on the ROMS side, we decided to show it only for FESOM.

15. "EPE" — this should really be APE (available potential energy) since eddies are deriving their KE from the APE of the /mean/ state (Vallis 2006 textbook).

Corrected.

16. Line 315: This is a nice demonstration that the flow is baroclincally unstable but is it unstable in the regions necessary to explain the eddy tracks in Figures 8,9? Some connection between the instability analysis and the eddy tracks is needed.

Eddy tracks are determined by both eddy generation and eddy movement. The latter as depicted in Figures 8,9 makes it hard to link the tracks to eddy generation associated with instability. Instead, we can compare the number of detected eddies (Figure 6a,b) with the maps of EKE (Figure 5) and the energy conversion rates (Figure 12). We added a subsection (Section 6.1) where we discuss this connection:
"Although baroclinic instability is the main driver of mesoscale eddy variability, the connection between eddy occurrences and EKE as well as the APE to EKE conversion rate (w'b') is very non-local. For one thing, the eddies form as the result of nonlinear evolution of baroclinic instability waves and jet meanders. For another, mean circulation transports and modifies all eddy-like features, moving them away from the sites where their 'seeds' originally appeared. As a result, the observed pattern of eddy occurrences (Figure 6a,b) differ from the EKE and w'b' distributions (Figures 5 and 12)...."

17. Lines 345, 346: I'm not sure there was any discussion of "EKE→MKE conversion associated with steepening isopycnals". Please describe this in more detail. The "energy backscatter" parameterizations being currently developed would be useful literature to reference (e.g.Jansen et al 2015)

We reformulated the last paragraph in the discussion, and hope to explain it better now. Thanks for pointing us to the study by Jansen et al. We also cite the papers by Juricke et al, which describe the implementation of the method by Jansen et al. into FESOM and its application to realistic test cases.

18. Discuss sensitivity of eddy maps to choice of 100m depth

We added a justification of the choice of the depth of 100 m:
"We decided to choose the depth of 100 m because both main water masses of the Fram Strait, AW and PW, are present at this depth (e.g. Wekerle et al., 2017, their Figure 9). In addition, the maximum relative vorticity is also located close to this depth for different regions of the Fram Strait (Figure 10)."

19. Figure 1: Showing a second panel with a larger region would be nice for readers not familiar with this place.

Thanks for this suggestion. We added a second panel in Figure 1 showing the Arctic Ocean and our study region.

20. Figure 4: It is more conventional to show logarithm axis scales instead of showing log10(quantity) on a linear scale. Why did you choose to not do that? Please mark 1/30 days since that's the frequency you chose to separate "mean" and "eddies"

As you suggested, we changed Figure 4 and now show logarithm axis scales. We also marked the frequency at 1/30 days.

---

## Author Comment (AC2) · 15 Jul 2020

**Reviewer 2**

The present paper reports results from a sensitivity study on the ability of two different numerical circulation models (ROMS and FESOM) in reproducing comparable eddy dynamic properties in terms of generation sites and propagation pathway in Fram Strait, a region characterized by a complex bathymetric configuration. Main difference between these models lies on their numerics and formulations, which includes numerical grid discretization, horizontal and vertical mesh resolution, parameterizations, and coverage (i.e. global vs. regional models).

The study is generally well written, and the analysis performed are appropriate. The figures are neat and clean. The discussion of the results is superficial. The authors should inspect more thoroughly the results presented. Nevertheless, the study stands as a good contribution, as it adds knowledge to the region. However, unfortunately it doesn't add new knowledge to the subject field whether in ocean modelling or in geophysical fluid dynamics. If it does, then perhaps it has not been clearly presented.

Find below few/minor points which the authors should address before its acceptance for publication.

We would like to thank you for your helpful comments. As you suggested, we improved the Discussion section. In particular, we added subsections discussing the connection between eddy occurrences and EKE (Section 6.1), the differences between models and observations (Section 6.2), and implications for contributing to future model development (Section 6.3). Below, we addressed your comments point by point.

The reference for ROMS in the introduction section may not be the most appropriate here!

In the introduction section, we added references to the papers by Shchepetkin and McWilliams (2005) and Budgell (2005).

Line 105 the sentence seems incomplete. In part it reads as "we also the Okubo-Weiss". Do you mean "we also used the Okubo-Weiss"?

Corrected.

Are there any criteria on the choice of the number of days taken as threshold to discard eddies with lifetime lesser than 3 days? Please note that the caption in Figure 10 and in other parts of the manuscript indicate a lifetime of 30 days. Is there any inconsistency??

We added a justification of the choice of the 3 days in section 2.3:
"We decided to use a threshold of 3 days mainly because the temporal resolution of the model output data is daily, and the eddy should form a track. This also helps to make sure that the eddies detected are real and not an over-detection due to uncertainties in the detection method. Eddies with a lifetime of at least three days are also required when computing the translation velocity needed to compute the eddy nonlinearity parameter, for which centred differences are used."
We analyzed all detected eddies (with lifetime>2 days) and discussed them in section 4.1, 4.2, 4.3 and 4.4. However, in sections 4.5 and 4.6 we decided to focus on long-lived eddies only. Analysis of pathways only makes sense for long-lived eddies. Otherwise the displacements are too short to be informative. In both sections, we made it clear that we only analyse "eddies with lifetime of more than 30 days".

Paragraph 130. What do you mean by eddy detected by experts?? Please make it clear.

Chaigneau et al. (2008) compared two methods of eddy detection, the 'winding-angle' method and Okubo-Weiss method. To validate their results, they asked oceanographic experts to detect eddies from sea level anomaly maps. Compared to the eddies detected by experts, the Okubo-Weiss method over-detected eddies. To avoid confusion, we just removed this sentence in the new version.

Did you use any filtering on the field of Okubo-Weiss parameter?

No, we do not use any filtering to compute the Okubo-Weiss parameter.

Perhaps the differences between the model maps shown in Figures 2 and 3 should be quantified to highlight geographical/spatial sites where the models converge and where they diverge (e.g: r=ROMS − FESOM for the parameters presented in Fig2 and Fig3).

Since the thermo-haline properties are not the main focus of this paper, we decided not to show the difference between ROMS and FESOM in T and S. Instead, we added contours of the 1°C and 2°C isotherms and the 34 and 35 isohalines, so that simulated T and S in both models can be more easily compared. Note that Figure 2 shows snapshots in time, so it would not make sense to plot the difference.

Under the Model assessment section, paragraph 155, the authors indicate the model simulates similar spatial distribution of the water masses. However, no T/S diagrams have been shown. Perhaps replacing the term "water-masses" with "thermo-haline properties"

would be more appropriate. The same is true wherever this term appears in this manuscript.

As suggested, we added T/S diagrams for ROMS and FESOM in Figure 3. The differences in the simulated thermohaline properties are better illustrated with it. Model differences are described now in more detail in section 3. We also exchanged the term "water-masses" with the term "thermo-haline properties" where it is appropriate.

Lines 170 – 173, is the difference only related to the effects of tides simulated in ROMS? Could not be also somehow related to differences in the surface forcing fields between the models?

This is correct. The surface atmospheric forcing as well plays a role. We added this in Section 3.

Lines 180 – 184: Are the eddy statistics computed in a Lagrangian or Eulerian frame of reference? Please clarify this important aspect.

We added this clarification in Section 4.2:
"Eddy properties such as their radius are determined at the locations where they are detected. In this sense, the eddy statistics are computed in a Lagrangian framework."

An important eddy property which could be included in this study is the eddy non-linearity parameter. This parameter would give a good insight on the eddy's coherence and ability to trap and transport material along their pathway of propagation.

Thanks for this suggestion. We added a new section (4.7) to analyse the eddy nonlinearity. Chelton et al. (2011) describe three different parameters to study eddy non-linearity, the advective nonlinearity parameter, the quasi-geostrophic nonlinearity parameter and the upper-layer thickness nonlinearity parameter. Here we decided to focus on the advective nonlinearity parameter, defined as the ratio of maximum rotational speed and translation speed.

Lines 248 – 249. Are the tides the only difference between the models? What about the vertical discretization of the water column??

We changed the sentence to:
"One of the many differences between the models..."

There are much more differences, which are discussed in Section 7. In this paragraph, we just speculate that tides are the main reason for the difference in eddy occurrences on the Yermak plateau in the models.

In section vertical extension and hydrographic properties: Are the values of vorticity in the eddy centers, a single point values or are averaged values within the eddies? Please make this aspect clear.

The relative vorticity in the water column is computed in the eddy center, and thus it is a single point value for every depth layer. We added in section 4.6: "...by calculating relative vorticity/f at the location of the eddy centres..."

Line 279: is the superscript number after (Figure 11a, b)1 meant to be there?

The superscript refers to a footnote.

---

## Author Comment (AC3) · 15 Jul 2020

**Comments by the Editor**

Thank you for this interesting manuscript. As you see, both reviewers have some constructive comments. I wanted to drop this editor comment so that you can consider the points below when preparing your final response and the revised version. My major criticism is that the discussion section is not well developed and must be improved.

*Thank you for your detailed comments. We improved the discussion section, and hope that it is better now.*

Here is a list of minor issues which must be clarified or written differently:

Li 25: "strongly turbulent": presence of mesoscale eddies does not make the oceanic conditions strongly turbulent. If this is supported by microstructure measurements, please use and cite; if not please choose a different wording (energetic?).

*Corrected. "Energetic" sounds much better.*

Li 33: do eddies lead to vertical eddy fluxes? I thought they would lead to lateral fluxes. Otherwise, please clarify the pathway from lateral to vertical.

*Eddies lead to both lateral and vertical fluxes, and the reason is that fluid particles move along isopycnals which are inclined. When eddies flatten isopycnals converting APE to EKE, lighter water moves upward, and denser downward along mean isopycnals. This is the main part of the GM parameterization. Therefore, eddies contribute to vertical heat flux. We reformulated point (2) in the introduction section:*
*"As shown by Hattermann et al. (2016), this region is characterised by negative values of vertical eddy temperature flux. Thus, eddy processes likely play an important role for the subduction of AW."*

Li 38. How is the MIZ "shaped" by eddies?

*We reformulated this sentence: "Eddies play an important role for sea ice-ocean interaction. The marginal ice zone is influenced by eddies..."*

Li 47-48: I do not think Teigen et al or Johannessen et al are the key references for the theory or dynamics related to barotropic instability or topographic steering/trapping of eddies, respectively.

We added the reference to Cushman-Roisin's book, and modified the sentence about barotropic instability: "... and eddies can be formed by receiving kinetic energy from the mean flow as shown for the Fram Strait by Teigen et al. 2011." We also added the reference to the study by Smith et al. (1984) which describes the topographic generation of an eddy in the EGC.

Li 92: Sundfjord et al (2017) is about a Svalbard fjord and I am not sure how it is relevant to the ability of the model to reproduce the slope boundary current.

We removed this reference.

Li 100: What are "a and b". Please clarify.

We added a description of parameters a and b.

Li 105: we also use

Corrected.

Li 117 and 118: please justify the choices of 3 day and 100 m depth

We added a justification of the choice of the 3 days and depth of 100 m in section 2.3: "We decided to use a threshold of 3 days mainly because the temporal resolution of the model output data is daily, and the eddy should form a track. This also helps to make sure that the eddies detected are real and not an overdetection due to uncertainties in the detection method. Eddies with a lifetime of at least three days are also required when computing the translation velocity needed to compute the eddy nonlinearity parameter, for which centred differences are used."
 "We decided to choose the depth of 100 m since both main water masses of the Fram Strait, AW and PW, are present at this depth (e.g. Wekerle et al., 2017, their Figure 9)."

Li 130: by experts? (please clarify)

We removed this sentence to avoid confusion.

Heading 2.4, remove one "and"

Corrected.

Li 145: these terms do not indicate instability but rather conversion from MKE to EKE and EPE to EKE (which can be related to the instabilities you mention).

We corrected this sentence: "... can be related to.."

Li 169-171: the syntax of frequencies and slopes are difficult to follow for a reader

We restructured the sentence, and hope that it is clearer now.

Li 193: did you introduce a stream function?

This was described in the methods section 2.3: "Eddy boundaries around each detected centre are determined by the outermost closed contour of the stream function field."

Li 200: Isn't this the Rossby number?

Right, this is the Rossby number. We used the Rossby number as an index for eddy intensity. We added a better description of the eddy intensity in section 4.3.

Li 229: Molloy

Corrected.

Li 293: The Ghaffari analysis is from a 2-layer model? I am not sure how this is directly comparable (at least you might want to point this out). About the instability of the slope current along the Lofoten escarpment, please see some recent conversion rate calculations similar to yours, using high resolution ROMS fields (Section 9, in a otherwise mooring observation paper): Ocean Sci., https://doi.org/10.5194/os-16-685-2020. Not that I authored this paper, so feel free to ignore this suggestion. However, the conversion rate fields (Fig 11 in both papers) are directly comparable.

Thanks a lot for hinting us to this recent study, it fits much better here and we thus cited it. It is also very valuable to be able to compare the magnitude of conversion rates to our estimates. We removed the reference to the paper by Ghaffari et al.

Li 309: Need a dot product before the buoyancy gradient?

Corrected.

Discussion section is not appropriate and must be improved. Also I note that the last paragraph (on providing information to develop GM type parameterization) is not really supported by your results or built upon them in a convincing way. Please improve this part or remove.

We improved the Discussion section. In particular, we added a new paragraph which discusses the differences between eddy occurrences and EKE. Moreover, we added a paragraph about "Implications for contributing to future model development". Here we also point to the energy backscatter scheme mentioned by Reviewer 1. We also improved the paragraph about the GM parameterization.

Opening paragraph of the Conclusions is not conclusions (or findings from your study), and could be integrated to discussion or removed.

We removed this paragraph.

Fram Strait: (If I'm not wrong) in the English usage, you should drop "the" in front of Fram Strait (except referring to a specific feature associated with Fram Strait, say, the Fram Strait circulation etc.) This must be corrected throughout, including the title.

Corrected.

Fig 4: I think it is not very meaningful to show the spectra of daily averaged speed (this is what you mean by absolute velocity?). It would be better to show the sum of spectra from u and v components (this corresponds to distribution of double the horizontal kinetic energy, or divide by two and call it HKE spectrum). In any case, you need units on y axis ((m/s)^2/(1/day) ?). It would be helpful with 95% confidence intervals on the spectra.

As suggested, we computed spectra for *u* and *v* separately and show now in Figure 4 the sum of the two spectra divided by 2. Now units are shown on the y-axis, and the 95% confidence intervals are indicated too.

---

## Author Response (AR2)

**Comment Reviewer 1:**

One minor comment: The response talks about address the Jansen et al parameterization and a paper by Juricke et al. Yet I could not find this in the PDF.

Thank you for pointing us to this mistake. We added the sentences about the Jansen et al. parameterization in section 6.3:
"A promising approach to reduce excessive dissipation in ocean models is the implementation of an energy backscatter scheme, which returns part of the over-dissipated energy back into the resolved flow (Jansen et al. 2015, Juricke et al. 2019). In a realistic application, Juricke et al. 2020 showed that eddy activity can be increased by a factor of 2, thereby also reducing biases in hydrography."

**Comments from the Editor:**

I find your response to the following points of referee#1 not thorough enough. The comment number (c#) refers to the reviewer 1's comment number, followed by my comment.

We would like to thank the editor for thoroughly going through the manuscript again. We improved the answers to the questions of Reviewer 1 as you suggested. We also addressed your minor comments.

c11: vertical extent. You are presenting vertical profiles (in the upper 500 m). Vertical extent typically refers to a thickness of the eddy based on a threshold. This is confusing. As is, in the response, you still do not explain how you define the extent. I do not think a definition of extent is necessary (as you do not discuss it). But I suggest you reword to vertical structure or profile.

It seems that the term "extent" is confusing, since we did not compute the vertical eddy thickness based on a threshold. In fact, we show the vertical profiles of relative vorticity, temperature and salinity computed for every eddy center location in 100 m depth in all model layers. Thus, the term "vertical structure" is more appropriate here. In the manuscript, we exchanged the term "extent" with "structure".

c13: You still don't explain or discuss why they are different. Also you do not seem to take any action in the manuscript in response to this comment.

Indeed, more cyclones are generated in FESOM in "FS Central South" than in ROMS (269 and 107 generated cyclones in FESOM and ROMS, respectively, during the years 2006-2009). Not only cyclones, but also anticyclones are more abundant in FESOM than in ROMS in this region. We added an explanation in Section 4.5:

"The difference in generated eddies in this region can be attributed to the different structure of the simulated mean flow and the temperature and salinity distribution (Figure 3), which is likely linked to the different model configurations (Table 1). The AW recirculation in FESOM is broader than in ROMS, so more eddies can be entrained with it. "
In fact, this is also reflected in a higher baroclinic energy conversion in this region in FESOM than in ROMS (Figure 12).

c14: This is not a good argument for not comparing (i.e., to reduce workload of the ROMS coauthors), especially when this is a model comparison paper.

As you suggested, we computed the term beta-$\partial_{xx}v$ for ROMS as well, and show the plot in the new Figure 13. As expected, both fields look relatively similar.

c18. You still do not discuss the sensitivity to 100 m. The new text only motivates the choice but does not discuss the sensitivity to the choice.

We added a discussion of the choice of 100 m depth for eddy detection (subsection 6.1):
"In this study, we chose the depth level of 100 m for eddy detection. Eddies present in the AW layer generally reach deeper than 100 m, so the eddy occurrence maps shown in Figure 6 are characteristic for deeper depths as well. In fact, an animation of daily averaged sections of velocity across Fram Strait, shown by Richter et al. 2018 (Movie S1 in the Supplement) using the same FESOM model output as this study, revealed that eddies in particular in the WSC region can reach very deep.
There may be some shallower eddies that we do not detect in 100 m depth. Shallow eddies have been observed in the Arctic Ocean Beaufort Gyre region, and are vertically confined by the strong stratification of the halocline (Zhao et al., 2014). Thus, using a shallower depth might cause us to overlook boundary current-origin eddies that do not penetrate the stratification below the mixed layer in the Basin. However, snapshots of relative vorticity close to the surface and at 100 m depth reveal a larger number of (small) eddies in 100 m depth, possibly due to strong stratification close to the surface (Figure not shown). A dedicated study of the vertical structure of eddies in Fram Strait as done by Zhao et al. (2015) for the Beaufort Gyre region is required."

The figure shown below depicts relative vorticity simulated in FESOM in different depths (30 m, 100 m and 300 m). It supports the above arguments. Most large eddies present in 100 m depth are also present in 300 m depth, yet, relative vorticity is weaker in 300 m than in 100 m depth. Some eddies present in 100 m depth are not present in 30 m depth, in particular the ones located between 81°N and 82°N. In general however, there is a large resemblance between the relative vorticity fields in 30 m, 100 m and 300 m depth.

[Figure]

Figure: Simulated relative vorticity divided by *f* at 30 m, 100 m and 300 m depth on Jan 1st, 2006 for FESOM.

li 304: shouldn't this be the opposite: salinity stratified and unstable in temperature?

In eastern Fram Strait where the ocean is dominated by warm AW, it is indeed temperature-stratified (e.g. von Appen et al. 2016, Hattermann et al. 2016). We removed the term "unstable in salinity", maybe this is confusing.

li 161: in describing how you obtained the EKE equation, did you miss one step (removing the MKE before multiplying by u')?

No, in fact the description of the EKE conservation equation is correct (please see Olbers et al. page 376). We changed the wording a bit, to make it clearer:

"An energy budget can be obtained by taking the time-average of the momentum equation in the Boussinesq approximation, expressing velocity $u = \bar{u} + u'$, multiplying the equation with $u'$, and time-averaging it again. This leads to a conservation equation for EKE (e.g. Olbers et al. 2012, chapter 12.2.1)."

Subtracting the energy equation of the mean flow from that of the total flow is not needed, because multiplication with $u'$ and averaging removes contribution from the mean automatically.

[revised manuscript text omitted]